# Molecular and anatomical organization of the dorsal raphe nucleus

Kee Wui Huang[1], Nicole E Ochandarena[1], Adrienne C Philson[1], Minsuk Hyun[1], Jaclyn E Birnbaum[1], Marcelo Cicconet[2], Bernardo L Sabatini[1]*

[1]Department of Neurobiology, Howard Hughes Medical Institute, Harvard Medical School, Boston, United States; [2]Image and Data Analysis Core, Harvard Medical School, Boston, United States

**Abstract** The dorsal raphe nucleus (DRN) is an important source of neuromodulators and has been implicated in a wide variety of behavioral and neurological disorders. The DRN is subdivided into distinct anatomical subregions comprised of multiple cell types, and its complex cellular organization has impeded efforts to investigate the distinct circuit and behavioral functions of its subdomains. Here we used single-cell RNA sequencing, in situ hybridization, anatomical tracing, and spatial correlation analysis to map the transcriptional and spatial profiles of cells from the mouse DRN. Our analysis of 39,411 single-cell transcriptomes revealed at least 18 distinct neuron subtypes and 5 serotonergic neuron subtypes with distinct molecular and anatomical properties, including a serotonergic neuron subtype that preferentially innervates the basal ganglia. Our study lays out the molecular organization of distinct serotonergic and non-serotonergic subsystems, and will facilitate the design of strategies for further dissection of the DRN and its diverse functions.
DOI: https://doi.org/10.7554/eLife.46464.001

## Introduction

The dorsal raphe nucleus (DRN) is a major source of neuromodulators in the central nervous system, and is the largest of the serotonergic nuclei, containing approximately a third of all serotonergic neurons (5-HT neurons) in the brain (*Hornung, 2010*). DRN 5-HT neurons send highly divergent projections that target many functionally distinct brain regions (*Azmitia and Segal, 1978*; *Muzerelle et al., 2016*; *Vertes, 1991*). Concordantly, diverse and at times conflicting functions have been proposed for DRN 5-HT neurons, including arousal (*Monti, 2010*), motor facilitation (*Jacobs and Fornal, 1997*), behavioral inhibition (*Miyazaki et al., 2012*), cognitive flexibility (*Matias et al., 2017*), value representation (*Bromberg-Martin et al., 2010*; *Cohen et al., 2015*; *Hayashi et al., 2015*; *Li et al., 2016*), motor learning (*Kawashima et al., 2016*), mood (*Cools et al., 2008*), and regulation of social interactions (*Dölen et al., 2013*; *Niederkofler et al., 2016*). Changes in DRN 5-HT neuron function have been implicated in various neuropsychiatric diseases, including major depressive disorder, bipolar disorder, schizophrenia, and obsessive-compulsive disorder, as well as movement disorders such as Parkinson's Disease (*Huot et al., 2011*; *Mahmood and Silverstone, 2001*; *Politis and Niccolini, 2015*; *Vaswani et al., 2003*). Serotonergic signaling pathways are also targets of widely-used therapeutic drugs, such as selective serotonin reuptake inhibitors (SSRIs) and atypical antipsychotics (*Meltzer and Massey, 2011*; *Vaswani et al., 2003*). However many of these drugs can acutely induce adverse side effects including anxiety and increased risk of suicidal behaviors (*Ferguson, 2001*; *Gartlehner et al., 2011*; *Vaswani et al., 2003*). This suggests these drugs affect multiple 5-HT pathways with distinct and contrasting effects on behavior and highlights the need for new approaches to target specific 5-HT neurons and their outputs (*Marcinkiewcz et al., 2016*; *Urban et al., 2016*). Understanding of the organization of the DRN and 5-HT neurons in functionally

*For correspondence:
bsabatini@hms.harvard.edu

Competing interests: The authors declare that no competing interests exist.

distinct subsystems may help explain conflicting conclusions for the role of serotonin in the etiology of mental disorders.

Neurons in the DRN are neurochemically and anatomically diverse – only 30% to 50% are serotonergic, and many release other neurotransmitters including GABA, glutamate, dopamine (DA), nitric oxide, and a variety of peptides (*Fu et al., 2010*). Some DRN neurons co-express genes that are traditionally used as markers for different neuronal cell types based on their neurotransmitter usage (e.g. *Gad1*, *Gad2*, *Slc17a6*, and *Slc17a8*) (*Fu et al., 2010*; *Okaty et al., 2015*). This presents a major challenge to attempts at separating cell types in the DRN using many genetic driver lines, which will label mixed populations in the DRN that have overlapping expression of these 'canonical' marker genes. Furthermore, these distinct neuronal cell types are spatially intermingled within the DRN and are difficult to separate anatomically.

Even bona fide DRN 5-HT neurons are molecularly, anatomically, and physiologically heterogeneous. Cytoarchitecturally, the DRN contains the 5-HT neuron clusters B6 and B7 (*Dahlström and Fuxe, 1964*; *Hornung, 2010*), and is further divided into subregions along all three major anatomical axes (*Hale and Lowry, 2011*). While these subregions can be roughly defined by the spatial clustering of 5-HT neurons, the density of 5-HT neurons varies gradually between the subregions preventing the delineation of clear boundaries. Although DRN 5-HT neurons share a common developmental lineage (*Deneris and Gaspar, 2018*; *Jensen et al., 2008*), DRN 5-HT neurons in different subregions express different G protein-coupled receptors and vesicular transporters, suggesting spatial clustering of functionally specialized neurons (*Calizo et al., 2011*; *Fernandez et al., 2016*; *Lowry et al., 2000*; *Templin et al., 2012*). Nevertheless, these anatomical subregions fail to capture many features of 5-HT neurons, as most of the variation does not show subregional specificity. Additionally, DRN 5-HT neurons vary in their axonal projection patterns, and the spatial distributions of these projection-defined subpopulations are broad and do not map cleanly to DRN subregions. This likely arises, in part, from the extensive collateralization of axons from individual 5-HT neurons (*Gagnon and Parent, 2014*; *Waselus et al., 2011*) – different 5-HT neurons are therefore likely to have overlapping axonal projection fields, even if they target distinct sets of regions to serve separate functions.

The heterogeneity of 5-HT neurons suggests that they may be organized into distinct subsystems. However, there is no consensus on the definition of these subsystems and only an incomplete understanding of correspondences between anatomically, molecularly, and functionally defined 5-HT neuron subtypes. Functional differences exist between projection-defined subpopulations of DRN 5-HT neurons (*Ren et al., 2018*), but it is unknown if these subpopulations map onto molecularly defined subtypes. Furthermore an unbiased census of DRN cell types and subtypes of 5-HT neurons is lacking, and presents a major obstacle in achieving a unified understanding of the diverse functions performed by these neurons.

Here we dissect the cellular composition of the DRN and relate molecular profiles of cell types to anatomical features. We used high-throughput single-cell RNA sequencing (scRNA-seq) to survey the cell types in and around the DRN and identified at least 5 distinct subtypes of 5-HT neurons, in addition to uncovering organizational principles for both neuronal and non-neuronal cell types based on gene expression profiles. Spatial information from in situ hybridization imaging datasets allowed us to infer the anatomical distribution of these DRN 5-HT neuron subtypes within the DRN. Molecularly defined subtypes of DRN 5-HT neurons are found in overlapping but different sets of anatomical subregions that do not match the spatial distribution of projection-defined subpopulations. Using viral retrograde tracing and multiplexed in situ hybridization, we characterized the expression of subtype-enriched genes in projection-defined neuronal subpopulations and found that the striatal-projecting 5-HT neuron subpopulation is a heterogeneous group comprised of at least 2 distinct molecular subtypes. Additionally, we defined a DRN 5-HT neuron subtype that is well-positioned to modulate basal ganglia circuits, based on its molecular and anatomical features. Our findings validate the feasibility of intersectional approaches to achieve the molecular and spatial specificity required to access distinct 5-HT neuron subsystems in vivo, and provide a resource to aid in the design of experimental strategies for the functional dissection of diverse DRN cell types and circuits.

## Results

### Cell type composition of the DRN by transcriptomic profiling

To survey the cell type composition of the DRN, we used the droplet-based inDrop platform (*Klein et al., 2015*; *Zilionis et al., 2017*) to perform transcriptomic profiling with single-cell resolution (*Figure 1A*). The DRN and surrounding areas were dissected from acute coronal brain sections spanning the rostro-caudal extent of the DRN, and dissociated cell suspensions were prepared by digesting the dissected tissue in a protease cocktail containing inhibitors of neural activity, transcription, and translation to reduce the effects of the dissociation procedure on gene expression (*Hrvatin et al., 2018*). Cells were sequenced to an average read depth of 48,805 reads per cell (median = 31,286 reads), and low-quality cells were removed prior to clustering (see Materials and methods). Aligned reads were filtered using unique molecular identifiers (UMIs) and quantified as UMI filtered mapped read (UMIFM) counts. Clustering using a graph-based algorithm and a shared-nearest-neighbors distance metric grouped yielded 17 major clusters (*Figure 1B*). These major clusters were identified and assigned to cell classes based their expression of previously-described marker gene combinations that were extensive cross-referenced with published datasets (*Marques et al., 2016*; *Okaty et al., 2015*; *Poulin et al., 2014*; *Saunders et al., 2018*; *Vanlandewijck et al., 2018*; *Zeisel et al., 2018*) (*Figure 1C*, *Supplementary file 1*). Smaller clusters containing putative doublets were identified based on co-expression of marker genes from different cell types and were removed manually. The processed dataset contained 39,411 cells from eight mice (4 male and 4 female, age range 8–10 weeks) with a median UMIFM count of 2,333 counts per cell (min. = 548; max. = 17,976; IQR = 1,903) and median gene detection rate of 1,034 genes per cell (min. = 216; max. = 5,765; IQR = 693). Cells from male and female mice were pooled for analysis as no major transcriptional sex differences were observed (*Figure 1—figure supplement 1*).

The majority of cells in the dataset were non-neuronal cells that included astrocytes, oligodendrocyte precursor cells (or polydendrocytes), differentiating and mature oligodendrocytes, ependymal cells of the cerebral aqueduct, lymphocytes, microglia, perivascular macrophages (pvMΦs), fibroblast-like or mesenchymal cells, endothelial cells, pericytes, and smooth muscle cells. Iterative subclustering identified subtypes of cells within each major non-neuronal class that included novel subpopulations – in addition to resolving different subtypes of endothelial cells (*Vanlandewijck et al., 2018*) and developmental stages of oligodendrocytes (*Marques et al., 2016*), we found multiple subtypes or states of astrocytes, oligodendrocytes, and ependymal cells. Ependymal cells shared expression of the histamine synthesis gene *Hdc*, but were further divided into distinct subtypes that differed in expression of cell adhesion and matrix-associated genes as well as signaling peptides such as *Pdyn* (*Figure 1—figure supplement 2A*). In situ hybridization (*ISH*) analyses of these differentially expressed genes revealed that these ependymal cell subtypes are topographically organized along major anatomical axes – *Pdyn*-expressing ependymal cells are restricted to ventromedial ventricular walls of the midbrain and hindbrain but are absent from the lateral and third ventricles in the forebrain (*Figure 1—figure supplement 2B*), suggesting zonation of functionally specialized ependymal cell subtypes throughout the ventricular system. Furthermore, the presence of these molecules used for both paracrine and synaptic communication suggests a locally specialized function of ependymal cells in signaling directly to neurons and other cells that express the appropriate G-protein coupled receptors, such as κ-opioid and histamine receptors.

Neurons (*n* = 2,041 cells), identified by expression of genes required for chemical synaptic transmission such as *Snap25*, clustered into 5 main classes that could be distinguished by their neurotransmitter usage. The 5 major neuronal classes (in decreasing order of abundance) are 5-HT, dopaminergic (DA), GABAergic, glutamatergic, and peptidergic neurons. Peptidergic neurons were depleted of genes for the synthesis and release of small molecule neurotransmitters (e.g. *Slc6a4*, *Slc6a3*, *Slc32a1*, *Slc17a6*, *Slc17a8*), but were highly enriched for the genes encoding neuropeptides such as *Cck*, *Cartpt*, *Ucn*, and *Postn*. Inspection of *ISHs* from the Allen Brain Atlas (*Lein et al., 2007*) indicated that these neurons were located in the Edinger-Westphal nucleus, which is adjacent to the DRN, confirming that our dissection region spanned most of the DRN along the anterior-posterior axis. Inspection of rhombomere-specific marker gene expression in the 5-HT neuron cluster showed a lack of markers for R2 (*Hoxa2*) and R3/5 (*Egr2*) derived neurons in contrast to the expression of R1 markers (*En1* and *En2*), indicating that our dissection, as intended, excluded most of the median raphe nucleus (*Okaty et al., 2015*).

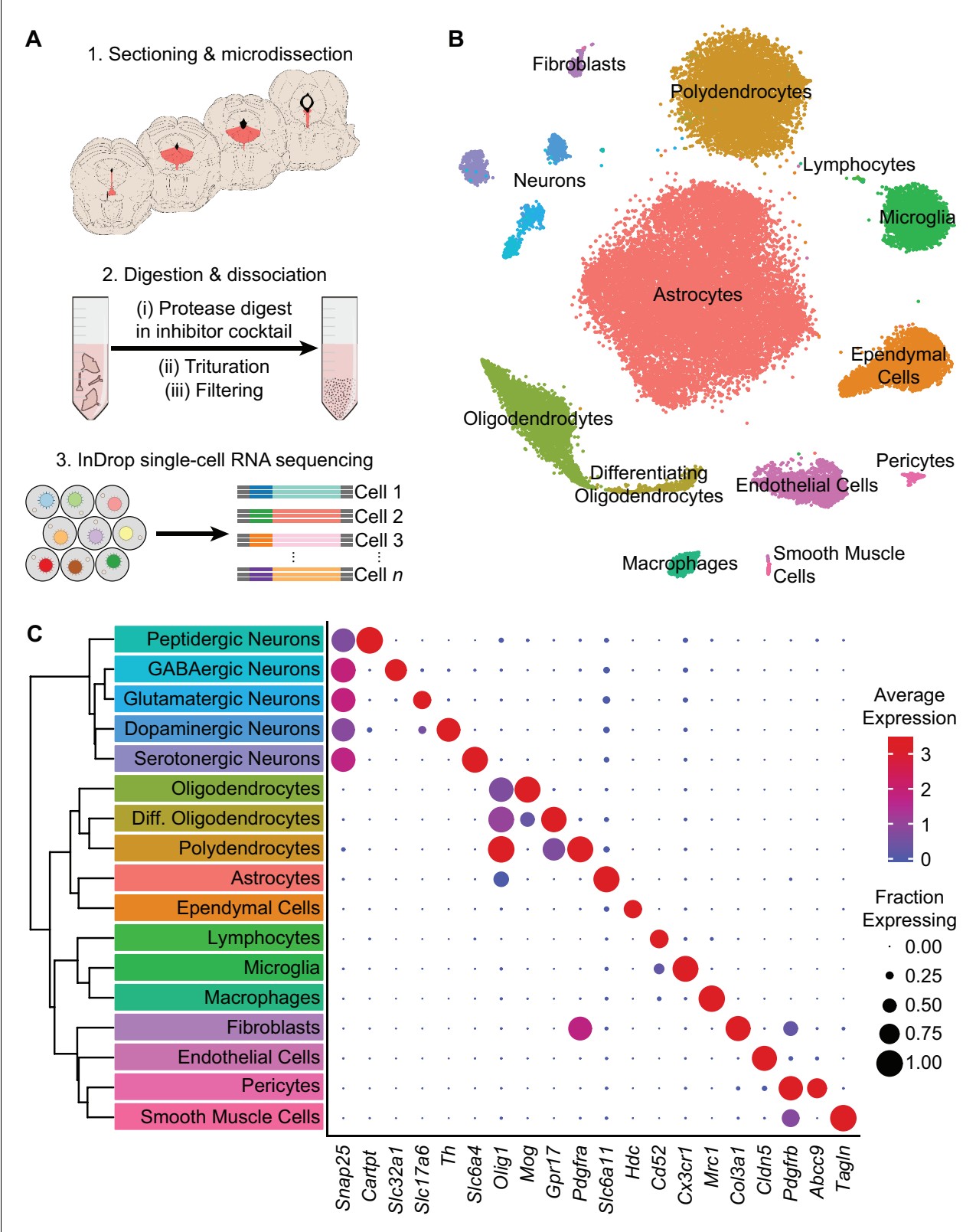

**Figure 1.** High-throughput single cell transcriptomic profiling of the DRN. (**A**) Schematic for scRNA-seq using the inDrop platform. (1) Tissue containing the DRN and surrounding regions was microdissected from acute coronal brain slices prepared from adult C57BL/6J mice. (2) Tissue chunks were digested in a cocktail of proteases and inhibitors of neuronal spiking, transcription, and translation. Digestion was followed by trituration and filtration to obtain a cell suspension. (3) Single cells were encapsulated using a droplet-based microfluidic device for cell barcoding and RNA capture. (**B**) t-SNE

*Figure 1 continued on next page*

*Figure 1 continued*

plot of the processed dataset containing 39,411 cells from eight animals. Cells are color-coded according to the cluster labels shown in (C). (C) Left: Dendrogram with cell class labels corresponding to clusters shown in (B). Right: Dot plot displaying expression of example genes used to identify each major cell class. The color of each dot (blue to red) indicates the average expression of each gene whereas the dot size indicates the fraction of cells expressing the gene. Additional genes used for identifying cell classes/types are listed in *Supplementary file 1*.

DOI: https://doi.org/10.7554/eLife.46464.002

The following figure supplements are available for figure 1:

**Figure supplement 1.** Comparison of cell type composition across sample batches and sex.

DOI: https://doi.org/10.7554/eLife.46464.003

**Figure supplement 2.** Ependymal cells lining the ventral cerebral aqueduct are of several distinct subtypes.

DOI: https://doi.org/10.7554/eLife.46464.004

## Serotonin receptor expression in neuronal and non-neuronal cell types

Analysis of single cell transcriptomic data allowed us to infer signaling between cell types based on expression of receptors and known ligands. As 5-HT can be released somatodendritically and from 5-HT axon terminals in the DRN (*Colgan et al., 2012*; *de Kock et al., 2006*; *Kaushalya et al., 2008*), we examined the expression patterns of 5-HT receptors in the dataset to infer local 5-HT signaling. Transcripts encoding 8 of the 15 5-HT receptors in the mouse genome were detected in our dataset (*Figure 2A*). Of these *Htr1a*, *Htr1b*, *Htr2c*, *Htr5b*, and *Htr7* were strongly expressed in different subsets of cells. The autoinhibitory $G_i$-coupled receptor *Htr1a* was expressed primarily in 5-HT neurons, whereas the $G_q$-coupled *Htr2c* receptor was expressed in both GABAergic and glutamatergic neurons (*Figure 2B*). Additionally, we unexpectedly observed expression of the $G_i$-coupled *Htr1b* receptor in both 5-HT neurons and pvMΦs of the DRN (*Figure 2C*). Examination of *Htr1b* expression in cortex, striatum, and ventral midbrain suggests that expression of this receptor in pvMΦs is unique to the DRN and its close surroundings (*Hrvatin et al., 2018*; *Saunders et al., 2018*; *Zeisel et al., 2018*). Additionally, the absence of abundant neuronal marker genes (e.g. *Snap25*) in pvMΦs indicates that the presence of *Htr1b* transcripts was unlikely to be a result of engulfment of neuronal debris containing mRNA (*Figure 2D*). *Htr5b*, which is a pseudogene in humans (*Grailhe et al., 2001*), was found in ependymal cells and a subset of 5-HT neurons. The $G_s$-coupled *Htr7* receptor was also found in a small subset of GABAergic and glutamatergic neurons.

## Molecularly distinct neuronal subtypes in the DRN

To further resolve the neuronal clusters into separate cell types or subtypes, we analyzed the neuronal clusters separately from the full dataset (see Materials and methods). Subclustering conservatively separated the 2,041 neurons into 18 distinct clusters, which we aggregated into 5 major groups based on similarities in their expression of genes, most notably those involved in the synthesis and release of different neurotransmitters (*Figure 3A*). These groups corresponded to the 5 major neuronal classes that were identified from clustering the full dataset, suggesting that neurotransmitter usage is a primary attribute for neuronal cell type identification. However, we also noted significant overlaps in the expression of genes whose loci are typically exploited for the generation of cell type-specific driver lines (*Figure 3B*, *Figure 3—figure supplement 1*). These genes included *Gad1* and *Gad2*, which encode GABA synthetic enzymes and are expressed in GABAergic neurons and a subset of both 5-HT and glutamatergic DRN neurons, as well as *Slc17a6*, which encodes a vesicular glutamate transporter and is expressed in most DRN glutamatergic and DA neurons. These overlaps in expression of 'canonical' marker genes are consistent with the release of multiple neurotransmitters by individual neurons, especially by monoaminergic neurons (*Granger et al., 2017*).

All of the major neuronal cell classes, with the exception of the peptidergic neurons, were comprised of several subclusters that each represent a distinct neuron subtype (*Figure 3B*); we found 5 5-HT neuron subtypes, 3 DA neuron subtypes, 5 glutamatergic neuron subtypes, 3 GABAergic neuron subtypes, and 1 cluster co-expressing markers for both GABAergic and glutamatergic neurotransmission (GABA/Glu cluster). To validate the existence of these mixed GABAergic/glutamatergic neurons, we performed multiplexed fluorescent in situ hybridization (FISH) for *Slc32a1* and *Slc17a6* as well as *Crhbp*, which was predicted to be strongly expressed by this cluster (*Figure 3C*). Cells co-expressing the three transcripts were found within the caudal-linear nucleus (CLi) and dorsal regions

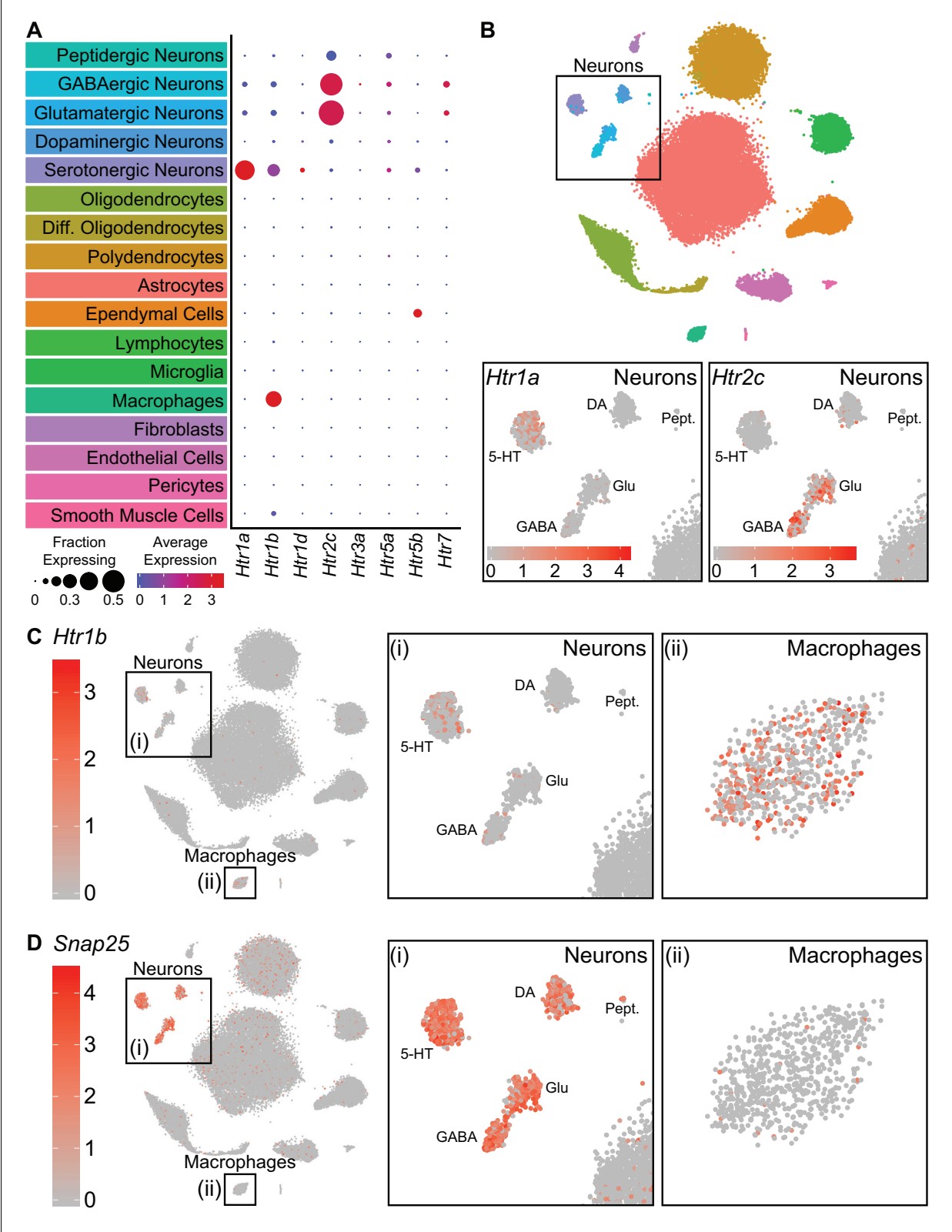

**Figure 2.** Serotonin receptors are expressed in both neurons and non-neuronal cells. (**A**) Dot plots showing expression of the serotonin receptors present in the scRNA-seq dataset in each major cell class. *Htr1a* is expressed specifically in 5-HT neurons, whereas *Htr1b* is expressed in both 5-HT neurons and perivascular macrophages. *Htr2c* is expressed in both GABAergic and glutamatergic neurons. *Htr5b* is expressed in ependymal cells and a subset of 5-HT neurons. *Htr7* is expressed in subsets of GABAergic and glutamatergic neurons. The remaining 5-HT receptors were detected at low
*Figure 2 continued on next page*

*Figure 2 continued*

levels. (**B**) t-SNE plots with the region containing neuronal clusters enlarged to show expression of *Htr1a* (bottom, left) or *Htr2c* (bottom, right) in individual cells. Cells are color-coded by expression of each transcript. Scale bars are in log-normalized UMI-filtered counts for the respective transcripts. (**C**) t-SNE plots with cells color-coded by their expression of *Htr1b*. Regions containing neuronal or perivascular macrophage clusters are enlarged. (**D**) t-SNE plots with cells color-coded by their expression of the neuron-specific transcript *Snap25*. Regions containing neuronal or perivascular macrophage clusters are enlarged. *Snap25* transcripts are abundant in all neuronal clusters but not in perivascular macrophages, suggesting that *Htr1b* transcripts in perivascular macrophages are unlikely to be a result of engulfment of neuronal debris.

DOI: https://doi.org/10.7554/eLife.46464.005

of the anterior DRN. Thus, this cluster, representing just 0.08% (32 of 39,411) of cells in the scRNA-seq dataset, corresponds to a bona fide and previously unrecognized neuronal cell type within the DRN that may be a caudal extension of the recently described GABA/Glu neuron group of the ventral midbrain (*Root et al., 2018*).

GABAergic and glutamatergic neurons shared the most similarities despite differences in neurotransmitter usage. We found that many subtypes of both GABAergic and glutamatergic neurons shared expression of genes that are enriched in the ventrolateral periaqueductal gray such as *Penk*, as well as the *Htr2c* receptor as described earlier. Within each major group, GABAergic and glutamatergic subtypes differed in their expression of a wide variety of genes that included those for many transcription factors, such as *Nkx2.2*, *Vsx2*, and *Satb1*, as well as peptides and transporters. Additionally, we found a glutamatergic neuron subtype that expressed the vesicular glutamate transporter *Slc17a8* (VGluT3), in contrast to all other DRN glutamatergic neuron subtypes that expressed *Slc17a6* (VGluT2). Consistent with previous descriptions, the *Slc17a8*-expressing glutamatergic subtype also expressed low levels of the transcription factor *Fev* (also known as *Pet1*) similar to 5-HT neurons, despite lacking genes required for the synthesis and release of 5-HT such as *Slc6a4* and *Tph2* (*Haugas et al., 2016*; *Hioki et al., 2010*).

Monoaminergic neurons were well separated into distinct DA and 5-HT classes. Although *Vip* was found in all 3 DA neuron subtypes, as previously reported for DRN and CLi DA neurons (*Dougalis et al., 2012*; *Poulin et al., 2018*; *Poulin et al., 2014*), *Vip* expression varied between the 3 subtypes. The $Vip^{low}$ DA-I subtype showed more similarities in gene expression with the GABA/Glu neurons of CLi as compared to the $Vip^{high}$ DA-III subtype, indicating a corresponding topographical organization of these DA neuron subtypes along the anterior-posterior axis from the CLi to DRN. $Vip^{low}$ DA-I subtype neurons also expressed nicotinic receptors including *Chrna4* and *Chrnb3* that are also found in GABA/Glu neurons, suggesting differences in nicotinic cholinergic activation of these DA neuron subtypes along the anterior-posterior axis and between DA neurons in CLi versus DRN. The DRN DA neurons lacking expression of nicotinic receptors separated into two subtypes, DA-II and DA-III, that differed in expression of genes related to neurotransmission including peptides, such as *Npw*, and the vesicular glutamate transporter *Slc17a6*, suggesting that DRN DA neurons are functionally heterogeneous.

## Genes differentially expressed between DRN 5-HT neuron subtypes

DRN 5-HT neurons (*n* = 704 cells) separated into five distinct subtypes. We constructed a cluster dendrogram using the averaged cluster gene expression to examine the molecular differences between these subtypes (*Figure 4A*, *Figure 4—figure supplement 1*). We found 333 differentially expressed genes (DE genes) by performing differential expression tests at each node on the cluster dendrogram at a 5% false discovery rate (Benjamini-Hochberg correction). Enrichment tests for gene ontology (GO) terms and analysis of DE genes with annotated functions showed that genes involved in neuronal development and identity specification, such as transcription factors, and genes involved in specifying neuronal connectivity, such as cell adhesion and axon guidance molecules, were over-represented in the transcripts differentiating the 5-HT subtypes (*Figure 4—figure supplement 2*). A large proportion of DE genes encoded for membrane proteins (GO:0016020 – membrane, 193 of 333 DE genes) with functions in neurotransmission and regulating intrinsic cellular excitability, such as neurotransmitter receptors, ion channels, vesicular transporters, neuropeptides, and other proteins involved in the synthesis or reuptake of neurotransmitters (*Figure 4A*).

The cluster dendrogram for 5-HT neuron subtypes showed that the smallest subtype cluster 5-HT-V (*n* = 7 cells) was the most transcriptionally distinct, was marked by expression of genes such as

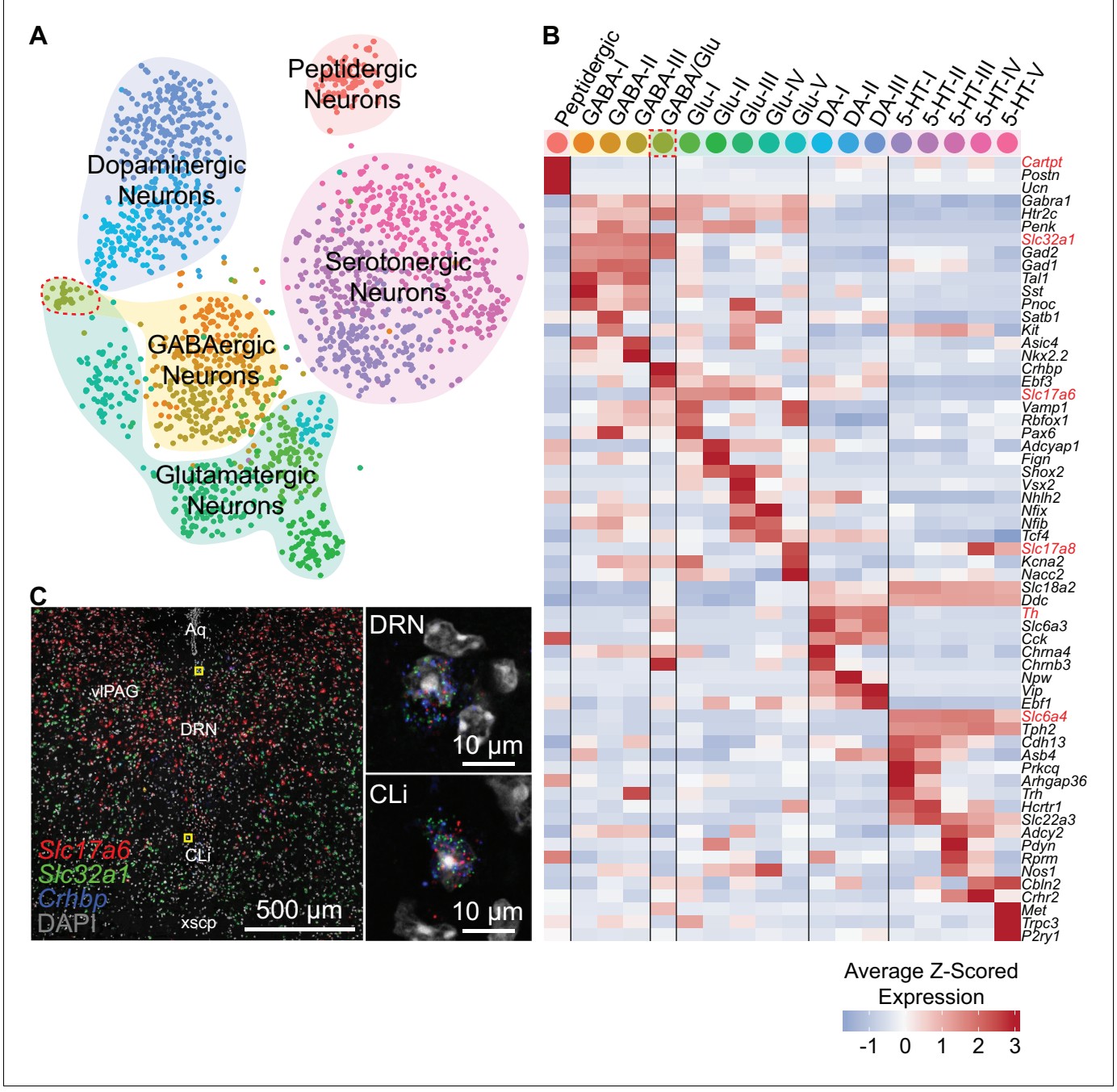

**Figure 3.** Neuron subtypes can be distinguished transcriptionally. (A) t-SNE plot of 2,041 neurons. The main neuron types were defined by the expression of genes necessary to make or release specific neurotransmitters or neuropeptides, and are highlighted by the shaded regions. Several subtypes of each class of neuron, designated by the cell color, can be contained in each shaded region. Color codes for subtype identities correspond to column headers in (B). The cluster outlined in red corresponds to cells expressing both glutamatergic and GABAergic markers. (B) Heatmap showing the cluster average Z-scored expression of selected genes that are enriched in each neuron subtype. Genes used to identify major neuronal cell types by their neurotransmitters are labeled in red. (C) Fluorescent in situ hybridization validating the existence of cells expressing both GABAergic and glutamatergic markers. Cells expressing *Slc32a1*, *Slc17a6*, and *Crhbp* were found in anterior regions of the DRN in the caudal linear nucleus (CLi) and dorsomedial DRN in close proximity to the cerebral aqueduct. The regions indicated by the yellow boxes are shown enlarged on the right.

DOI: https://doi.org/10.7554/eLife.46464.006

The following figure supplement is available for figure 3:

**Figure supplement 1.** Average log-scaled expression of genes differentially expressed between neuron subtypes.

DOI: https://doi.org/10.7554/eLife.46464.007

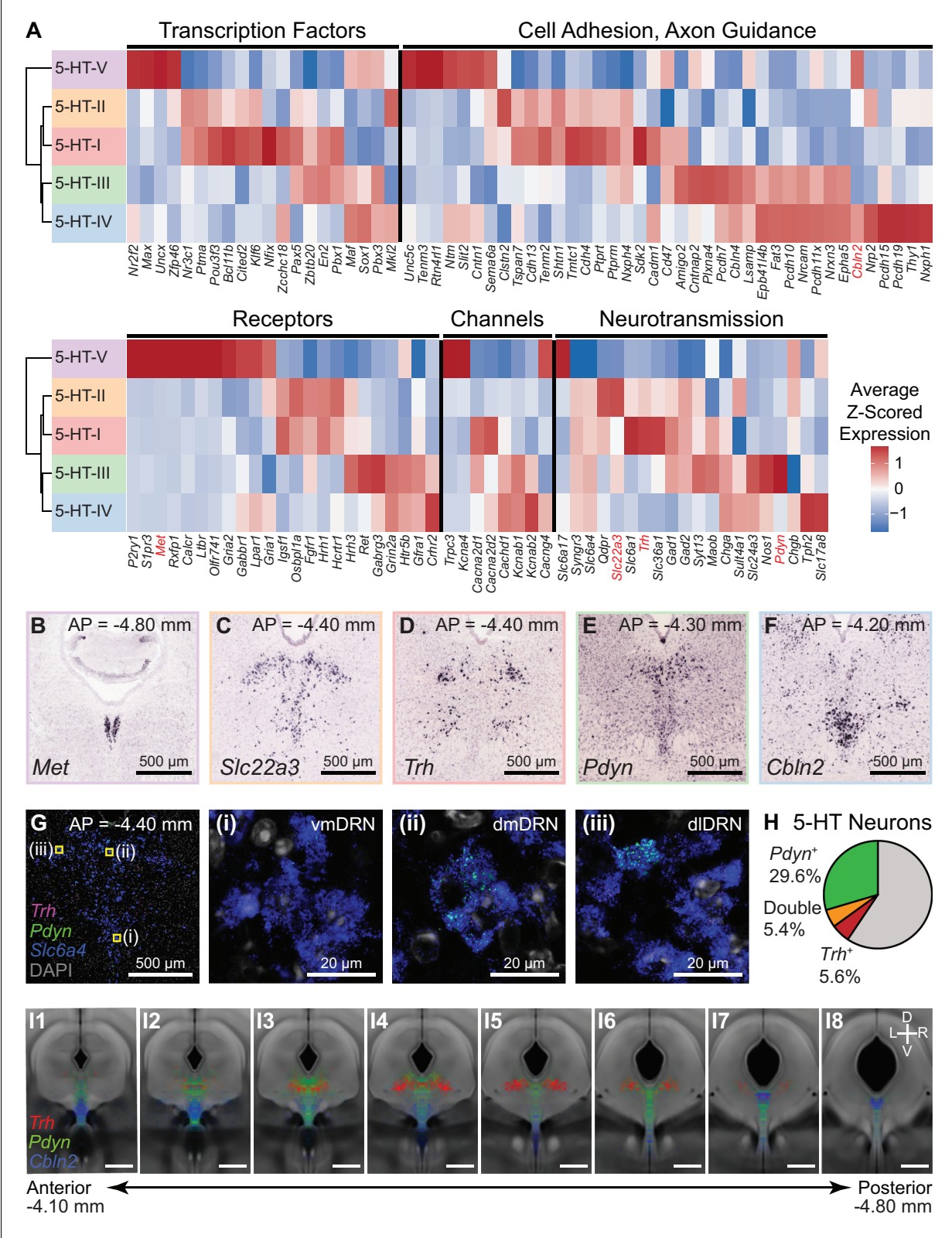

**Figure 4.** Characterization of genes differentially expressed between serotonergic neuron subtypes. (**A**) Z-scored expression heatmaps of selected genes with functional annotations that are differentially expressed between DRN 5-HT neuron subtypes. A total of 333 differentially expressed (DE) genes were found (5% false discovery rate, Benjamini-Hochberg correction). A large proportion of DE genes encode for cell surface and transmembrane proteins. Genes shown are categorized into five groups: transcription factors, cell adhesion and axon guidance molecules,

*Figure 4 continued on next page*

*Figure 4 continued*

transmembrane receptors, ion channels, and neurotransmission-related genes. (B-F) In situ hybridizations from the Allen Brain Atlas showing the expression patterns of selected DE genes enriched in each of the five 5-HT neuron subtypes. Many DE genes are not expressed uniformly throughout the DRN and are instead spatially heterogenous along the anterior-posterior, medial-lateral, and dorsal-ventral axes, suggesting that 5-HT neuron subtypes are differentially distributed across spatial domains. (G) Confocal images from a representative DRN coronal section (AP = −4.40 mm) stained for mRNA transcripts of *Trh* (magenta), *Pdyn* (green), and *Slc6a4* (blue) using RNAscope single molecule fluorescent in situ hybridization (*FISH*). Regions marked in yellow are enlarged in images (i)-(iii) and show differential expression of *Trh* and *Pdyn* in three DRN subregions. *Trh* and *Pdyn* expression is low in the ventromedial DRN (vmDRN), whereas 5-HT neurons expressing high levels of *Trh* or *Pdyn* can be found in the dorsomedial DRN (dmDRN) and dorsolateral DRN (dlDRN). Cells expressing both *Pdyn* and *Trh* were occasionally observed in the regions bordering dmDRN and dlDRN. (H) Proportion of 5-HT neurons ($Slc6a4^+$) containing *Pdyn* or *Trh* transcripts (minimum of 3 puncta). *Trh* transcripts were detected in a small proportion of $Pdyn^+$ 5-HT neurons. (I) Probability distribution maps of $Slc6a4^+$ 5-HT neurons containing transcripts for *Pdyn* (green), *Trh* (red), or *Cbln2* (blue) across multiple *FISH* experiments. Cells of each category were separately mapped to a reference atlas and counted in 50 μm x 50 μm bins. Bin counts were normalized to the total cell count for each category to obtain the probability maps that are displayed as heatmaps overlaid on the average coronal Nissl template from the Allen Brain Atlas. *Pdyn*-expressing 5-HT neurons are enriched along the midline, whereas *Trh*-expressing 5-HT neurons are enriched in dorsolateral regions. *Cbln2*-expressing 5-HT neurons are distributed more ventrally, and are also found in the posterior B6 tail of the DRN (panel I7). Positions of $Slc6a4^-$ cells are not shown. Scale bars: 500 μm, unless otherwise stated.

DOI: https://doi.org/10.7554/eLife.46464.008

The following figure supplements are available for figure 4:

**Figure supplement 1.** Average log-scaled expression of genes differentially expressed between 5-HT neuron subtypes.
DOI: https://doi.org/10.7554/eLife.46464.009
**Figure supplement 2.** Enriched gene ontology terms associated with genes differentially expressed between 5-HT neuron subtypes.
DOI: https://doi.org/10.7554/eLife.46464.010
**Figure supplement 3.** Additional validation of differential gene expression between 5-HT neuron subtypes using *FISH*.
DOI: https://doi.org/10.7554/eLife.46464.011

*Met*, *P2ry1*, and *Trpc3* (*Figure 4—figure supplement 3C*), and lacked expression of genes for GABA co-transmission. Inspection of *ISHs* probing these genes suggests that 5-HT-V subtype neurons are located in the posterior B6 tail of the DRN (*Figure 4B*).

The remaining four subtypes separated into two groups: the first was comprised of subtypes 5-HT-I (*n* = 178 cells) and 5-HT-II (*n* = 186 cells) that had enriched expression of genes for GABAergic neurotransmission such as *Gad1* and *Slc6a1*, and the calcium-independent protein kinase *Prkcq*. The second group was comprised of subtypes 5-HT-III (*n* = 125 cells) and 5-HT-IV (*n* = 208 cells) that were enriched in genes such as *Adcy2*, which was not expressed in the 5-HT-I/II group. The 5-HT-III/IV group also expressed different cell adhesion and axon guidance molecules from the 5-HT-I/II group, suggesting differences in both cell location and axonal projections between them. However, the expression of several genes in 5-HT-III appeared to be intermediate between 5-HT-I/II and 5-HT-IV – while the 5-HT-III/IV group showed reduced expression of genes for GABAergic neurotransmission relative to subtypes 5-HT-I/II, several genes that were enriched in 5-HT-III over 5-HT-IV were also expressed in subtypes 5-HT-I or 5-HT-II, including *Pax5*, *Nxph4*, *Gad1*, and *Gad2*. Although the 5-HT-III/IV group expressed genes for glutamatergic transmission, subtype 5-HT-IV expressed higher levels of the vesicular glutamate transporter *Slc17a8* compared to 5-HT-III. 5-HT-III was also enriched in several genes such as *Pdyn*, *Ret*, *Plxna4*, and *Hrh3* compared to the other subtypes. The features shared between 5-HT-III with both 5-HT-IV and the 5-HT-I/II group are suggestive of gradients in gene expression along orthogonal anatomical axes on which the subtypes are differentially distributed.

*ISHs* for genes enriched in these four subtypes revealed expression patterns that were consistent with the hypothesis that the two groups 5-HT-I/II and 5-HT-III/IV largely separated along the medial-lateral axis, with the added separation of 5-HT-IV from the other three subtypes along the dorsal-ventral axis. Genes enriched in subtypes 5-HT-I and 5-HT-II were expressed in lateral parts of the DRN (*Figure 4C–D*), whereas genes enriched in subtypes 5-HT-III and 5-HT-IV showed enrichment in medial parts of the DRN along the midline (*Figure 4E–F*). Additionally, the genes that were differentially expressed between subtypes 5-HT-III and 5-HT-IV showed graded expression along the dorsal-ventral axis along the midline. Multiplexed fluorescent *ISH* (*FISH*) further validated the differential expression of several subtype-enriched genes between 5-HT neurons in different spatial subregions of the DRN (*Figure 4G–I*, *Figure 4—figure supplement 3*).

## Spatial distribution of 5-HT neuron subtypes in the DRN

Anatomical features are lost during the dissociation of tissue for droplet-based scRNA-seq techniques. *ISH* against a small set of marker genes expressed exclusively in one cell type can be used to localize the putative location of cell types in complex tissue. However, this approach ignores the full information contained in the many dozens or hundreds of genes whose expression differs in a graded manner across different cellular subtypes. In order to exploit information from many genes differentially expressed across 5-HT neuron subtypes to infer the spatial distribution of transcriptionally defined subtypes, we utilized large imaging datasets from the Allen Brain Atlas, which contain the brain-wide spatial expression profiles of nearly every gene. We built a matrix of spatially binned gene expression for DE genes using images of coronal *ISH* spanning the DRN registered to the Allen Brain Atlas Common Coordinate Framework (see Materials and methods, *Figure 5—figure supplement 1A–B*, and *Supplementary file 2*). We first used the spatial distribution of *Tph2* expression to define spatial bins (voxels) within the Common Coordinate Framework that contain DRN 5-HT neurons. Spatial *ISH* expression data was additionally filtered to retain only spatially variable genes within the region defined by the *Tph2* expression mask (*Figure 5—figure supplement 1C*). Subsequently, correlation coefficients between the normalized expression scores for 112 DE genes for each of these voxels and each 5-HT neuron subtype cluster were calculated (*Figure 5—figure supplement 1D*, *Figure 5—figure supplement 2*). Collections of voxels with high positive correlation coefficients represented the inferred spatial distribution of each subtype within the DRN (*Figure 5*). Missing data in the Allen Brain Atlas dataset as well as artifacts apparently introduced in tissue processing that hampered alignment to the Common Coordinate Framework prevented us from analyzing the full 333 DE genes.

Correlation maps showed that the 5-HT neuron subtypes identified from the scRNA-seq dataset have distinct but overlapping spatial distributions spread along all three major anatomical axes. Most subtypes were not confined to a specific DRN subregion, with the exception of subtype 5-HT-V (*Figure 5*, purple) that showed a tight distribution within the dorsomedial posterior B6 tail of the DRN. The remaining four 5-HT neuron subtypes were found within the larger anterior B7 part of the DRN, and were segregated along both the medial-lateral axis and the dorsal-ventral axis. Subtypes 5-HT-I and 5-HT-II showed the greatest similarities in spatial distribution – both subtypes occupied the dorsolateral subregions in the posterior half of the DRN where they are likely spatially intermingled, consistent with their higher expression of GABAergic transmission genes *Gad1* and *Gad2*. In the more anterior DRN where the lateral wings are absent, subtypes 5-HT-I (*Figure 5*, red) and 5-HT-II (*Figure 5*, orange) were also be found within the dorsomedial subregion. This overlaps with the distribution of the *Pdyn*-enriched subtype 5-HT-III (*Figure 5*, green), which showed the highest correlations with voxels in the dorsomedial part of the DRN in its anterior half. Both subtypes 5-HT-III and 5-HT-IV (*Figure 5*, blue) showed higher correlations with voxels along the midline, but were separated along the dorsal-ventral axis with the *Slc17a8*-enriched 5-HT-IV subtype occupying most of the ventral DRN. Subtype 5-HT-IV was also most evenly distributed along the anterior-posterior axis compared to the other subtypes.

## 5-HT neuron subtype distributions coincide with different projection-defined subpopulations

The nuclei of the basal ganglia are major targets of DRN 5-HT neurons, with approximately one-third of all DRN 5-HT neurons innervating the striatum (Str) (*Steinbusch et al., 1981*). In addition to the extensive anatomical connectivity between the basal ganglia and DRN, both structures have motor and cognitive functions, and have been implicated in disorders that affect motor control and motivation such as impulsivity, obsessive-compulsive disorder, and major depressive disorder (*Di Matteo et al., 2008*; *Miguelez et al., 2014*; *Pauls et al., 2014*; *Schilman et al., 2010*). Although the DRN has many shared functions with the basal ganglia, DRN neurons are also involved in modulation of sensory pathways, limbic and neuroendocrine systems, and brainstem motor nuclei, and it has been proposed that anatomically segregated subsets of DRN 5-HT neurons form separate efferent pathways to perform these different functions (*Hale and Lowry, 2011*; *Lowry, 2002*; *Muzerelle et al., 2016*; *Waselus et al., 2011*). Therefore to compare the distribution of the 5-HT neuron subtypes from our scRNA-seq dataset with those of projection-defined neuron subpopulations in the DRN, we performed retrograde tracing from several nuclei of the basal ganglia, structures connected to basal

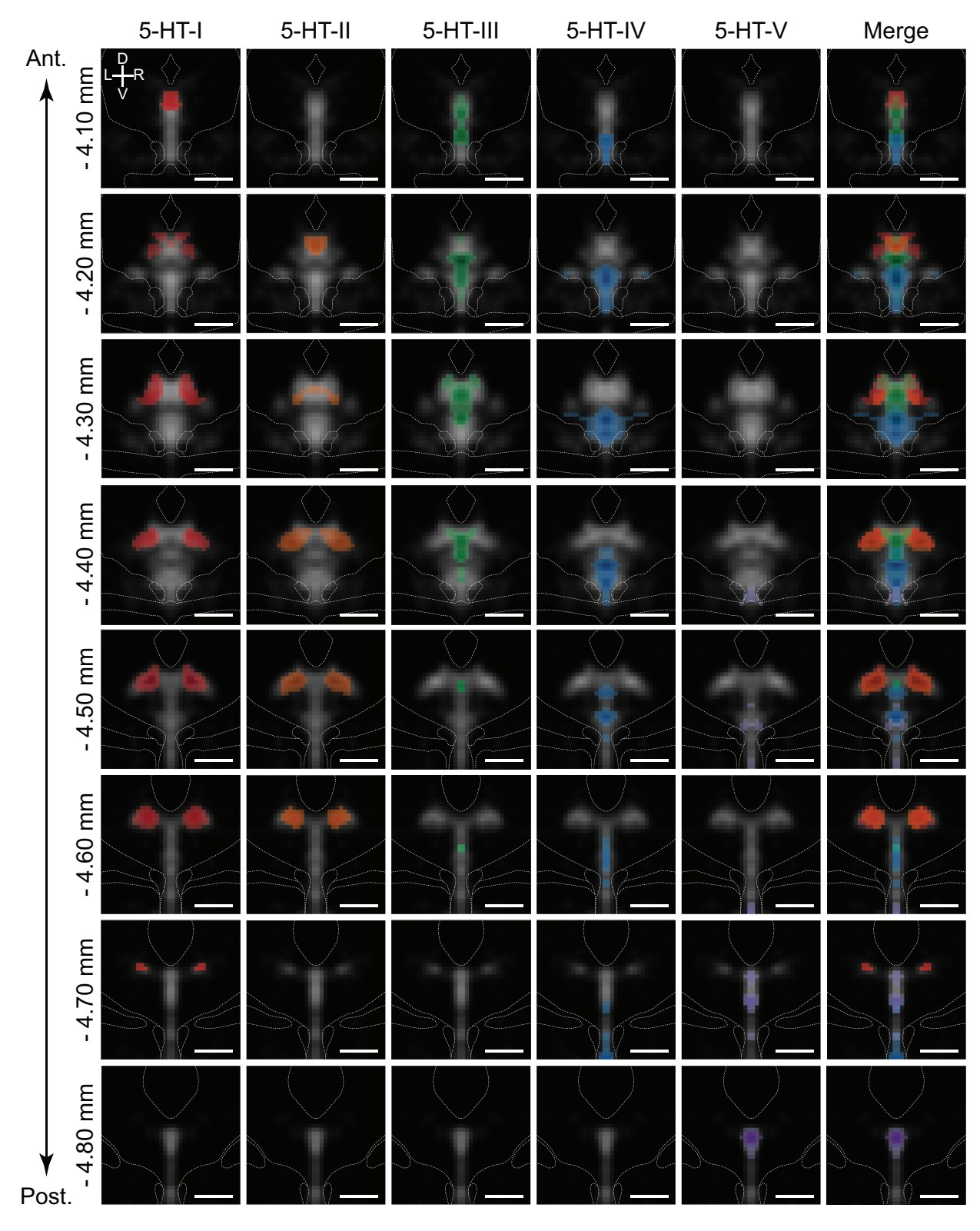

**Figure 5.** 5-HT neuron subtypes are distributed in distinct but overlapping spatial domains. Inferred spatial distributions of 5-HT neuron subtypes displayed as a map of correlation coefficients between profiles of gene expression of each subtype and each spatial bin. Each column shows the correlation map for each 5-HT neuron subtype at different coronal sections along the anterior-posterior axis of the DRN (rows). Only bins with positive correlation coefficients greater than 0.1 are shown, and maps were smoothed using a 2D Gaussian filter (1SD) for display. Correlation maps are overlaid

*Figure 5 continued on next page*

*Figure 5 continued*
on the averaged *Tph2* expression (grayscale, based on *ISH* data from the Allen Brain Atlas). Anatomical landmarks visible in the averaged Nissl template, such as the cerebral aqueduct and fiber tracts, are outlined and overlaid on each coronal image. Correlation maps for subtypes 5-HT-I and 5-HT-II suggest that they are largely intermingled in the dorsolateral DRN, with some overlap in the anterior DRN with 5-HT-III in the dorsomedial DRN. 5-HT-IV was enriched in the ventral regions of the DRN, while 5-HT-V was highly restricted to the posterior B6 tail of the DRN that was highly enriched in expression of the 5-HT-V enriched gene *Met*. Scale bars: 500 µm.
DOI: https://doi.org/10.7554/eLife.46464.012
The following figure supplements are available for figure 5:

**Figure supplement 1.** Processing of *ISH* images for spatial correlation analysis.
DOI: https://doi.org/10.7554/eLife.46464.013
**Figure supplement 2.** Correlation coefficient maps for DRN 5-HT neuron subtypes.
DOI: https://doi.org/10.7554/eLife.46464.014

ganglia circuits such as frontal and prefrontal cortical regions, as well as more functionally and anatomically distant structures that include thalamic sensory nuclei, such as the dorsal lateral geniculate nucleus (dLGN).

To map the distribution of projection-defined DRN neuron subpopulations, we performed retrograde tracing using glycoprotein-deleted rabies viruses (RbV) injected into brain regions known to receive inputs from DRN 5-HT neurons (*Figure 6A*). Cells infected via their axons by RbV were fluorescent, and their positions were manually registered onto an averaged Nissl coronal template from the Allen Brain Atlas (*Figure 6B–C*). Cells were counted in 50 × 50 µm bins, and bin counts were normalized by the total cell count to obtain the probability distributions for each projection-defined subpopulation. Probability distributions are displayed as heatmaps overlaid on reference template images to compare their distributions.

Most projection-defined subpopulations were located either ipsilateral to the injection site or along the midline, with the exception of the habenula-projecting neurons which were found in both ipsilateral and contralateral lateral wings of the DRN (*Figure 6—figure supplement 1*). Within the cortical-projecting group, frontal- and prefrontal-projecting subpopulations were differentially distributed along the anterior-posterior axis of the DRN – M1-projecting neurons were relatively confined to the anterior half of the DRN, whereas mPFC-projecting neurons were found in both anterior and posterior halves. Consistent with previous reports (*Ren et al., 2018*; *Waterhouse et al., 1986*), we found that most subcortical-projecting neurons were located in the dorsal DRN, whereas most cortical-projecting neurons were located in the ventral DRN. Although there was a crude separation of cortical- and subcortical-projecting subpopulations along the dorsal-ventral axis, there were considerable overlaps between their distributions, preventing the delineation of a clear boundary between cortical- and subcortical-projecting groups. The Str-projecting and M1-projecting subpopulations were largely segregated into the dorsal and ventral DRN respectively. However, there were overlaps in their distributions at intermediate levels – both Str-projecting and M1-projecting neurons were found in the ventrolateral DRN (outlined in cyan in *Figure 6D* column 3) and dorsal border of the ventromedial DRN. Double retrograde tracing experiments also showed that DRN neurons innervating both Str and M1 were often found in these regions (*Figure 6—figure supplement 2*).

Projection-defined subpopulations that innervated different subcortical targets, such as the Str-projecting versus dLGN-projecting subpopulations, were spatially segregated within the DRN (*Figure 6D*). dLGN-projecting neurons were mostly located in dorsolateral subregions at more posterior levels, whereas Str-projecting neurons were mostly in the dorsomedial subregion at more anterior levels. The distributions of subpopulations innervating related structures within the basal ganglia were more similar (*Figure 6E*) and, while the subpopulations innervating the external globus pallidus (GPe) and Str were most similar, the subpopulations innervating the Str and nucleus accumbens (NAc, also called the ventral striatum) showed a larger difference. This difference was largely driven by the presence of NAc-projecting neurons in posterior parts of the DRN where mPFC-projecting neurons were also located, whereas Str-projecting DRN neurons were relatively confined to the anterior half of the DRN (*Figure 6—figure supplement 3*). This is also consistent with differences in the function of Str and NAc and their relation to functional differences between the anterior and posterior DRN (*Lowry, 2002*; *Van Bockstaele et al., 1993*).

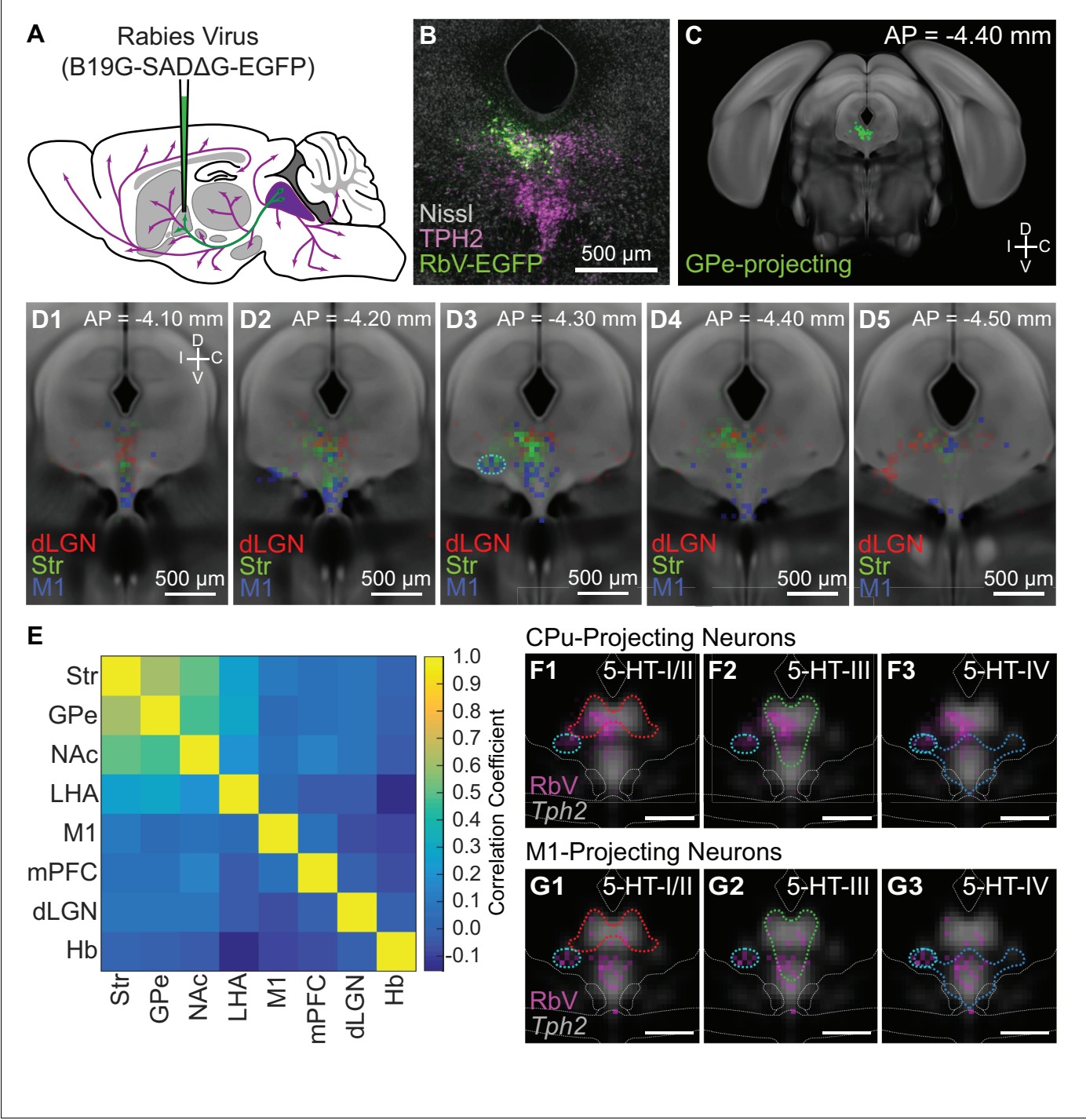

**Figure 6.** Projection-defined DRN neuron subpopulations overlap with multiple 5-HT subtypes. (**A**) Schematic of retrograde tracing experiments in which unpseudotyped rabies viruses (SADΔG B19 stain) were used to infect and label neurons whose axons innervate the virus-injected target region (GPe in this example). (**B**) Image of a coronal section of the DRN showing rabies-labeled, GPe-projecting neurons. Brains were fixed 7 days post-injection for histology, and 5-HT neurons in the DRN were visualized using immunohistochemistry against TPH2. Sections were counterstained using a fluorescent Nissl stain (Neurotrace). (**C**) The location of all rabies-labeled cells within the DRN and vlPAG, such as the GPe-projecting cells in (**B**), were manually registered onto a coronal template derived from the averaged Allen Brain Atlas coronal Nissl stain. Labeled neurons outside of this region of interest, such as the PPN and dorsal PAG, were not included. (**D1-5**) Distributions of Str-projecting (green), M1-projecting (blue), and dLGN-projecting (red) subpopulations, displayed as a probability density map (50 μm x 50 μm bins) overlaid on the coronal template. Coronal sections shown are centered in panel D3 at the maximum density of the Str-projecting population. Str- and M1-projecting populations were largely separated along the dorsal-ventral axis, with some overlap. Co-labeled neurons projecting to both Str and M1 were typically found in the ventrolateral DRN, outlined in cyan

*Figure 6 continued on next page*

*Figure 6 continued*

in D3, and the ventromedial DRN. (**E**) Pairwise correlations between the distributions of projection-defined subpopulations. Correlations between the Str-projecting subpopulation were higher with other subpopulations that also innervate nuclei of the basal ganglia, compared with subpopulations innervating functionally distinct targets such as M1 or dLGN. (**F-G**) Overlap between the distributions of neurons innervating Str (**F**) or M1 (**G**) and the 5-HT neuron subtype distributions at AP = −4.30 mm. Distributions of each projection-defined subpopulation are represented as probability maps (magenta) overlaid on the averaged *Tph2* expression. Subtype distributions are represented as color-coded outlines corresponding to the colors used in *Figure 4* for each subtype. Str-projecting neurons largely fell into regions occupied by both 5-HT-III and 5-HT-IV subtypes, suggesting molecular heterogeneity within this projection-defined population. M1-projecting neurons were more confined to the ventral DRN, overlapping with the distribution of the 5-HT-IV subtype. Scale bars: 500 µm.

DOI: https://doi.org/10.7554/eLife.46464.015

The following figure supplements are available for figure 6:

**Figure supplement 1.** Spatial distributions of projection-defined subpopulations.

DOI: https://doi.org/10.7554/eLife.46464.016

**Figure supplement 2.** Ventrolateral DRN neurons send axon collaterals to both striatum and motor cortex.

DOI: https://doi.org/10.7554/eLife.46464.017

**Figure supplement 3.** Caudal DRN B7 neurons innervate ventral striatum and prefrontal cortex.

DOI: https://doi.org/10.7554/eLife.46464.018

Differential expression of many axon guidance and cell adhesion molecules between 5-HT neuron subtypes predicts that 5-HT subtypes innervate different brain regions. Two models for how these molecularly-defined subtypes relate to projection-defined subpopulations that would be consistent with this hypothesis are (i) 5-HT neuron subtypes have non-overlapping axonal projection fields and thus a projection-defined subpopulation is comprised of a single subtype and is contained within the subtype's distribution, or (ii) different 5-HT neuron subtypes can innervate the same structure and thus projection-defined subpopulations can be comprised of one or more subtypes. To predict the 5-HT neuron subtypes present in different projection-defined neuron subpopulations, we compared the distributions of projection-defined subpopulations with the inferred 5-HT neuron subtype distributions.

It has been reported that cortical-projecting subpopulations of DRN 5-HT neurons express *Slc17a8*, in agreement with the extensive overlap between the spatial distribution of cortical-projecting neurons and *Slc17a8* expression (*Ren et al., 2018*). This suggests that cortical-projecting subpopulations should correspond to subtype 5-HT-IV from the scRNA-seq dataset, which is *Slc17a8*-expressing and distributed to the ventral DRN. Additionally, we would predict that the Str-projecting subpopulation corresponds to the subtype 5-HT-III, since their distributions overlap extensively in the dorsomedial DRN. Consistent with our hypothesis that 5-HT neuron subtypes innervate different brain regions, we found that the inferred distributions of each 5-HT neuron subtype coincides with a different set of projection-defined subpopulations – the distribution of subtype 5-HT-III was most similar to the distribution of Str-projecting neurons (*Figure 6F*), whereas the distribution of subtype 5-HT-IV was more similar to that of M1-projecting neurons (*Figure 6G*). However, neither Str-projecting nor M1-projecting subpopulations were fully contained within the distribution of a single 5-HT neuron subtype. Str-projecting neurons in the ventrolateral DRN (marked cyan in *Figure 6D,F,G*), where neurons innervating both Str and M1 are located, fell within the distribution of subtype 5-HT-IV but not of subtype 5-HT-III. This suggests that Str is a target of multiple molecular subtypes of 5-HT neurons, and supports the model that some structures are innervated by multiple 5-HT neuron subtypes due to overlaps in the axonal projection fields of different subtypes.

## Striatal-projecting 5-HT neurons are transcriptionally heterogeneous

To determine the molecular identities of individual projection-defined DRN neurons, we combined RbV-mediated retrograde tracing with multiplexed *FISH* for genes enriched in different 5-HT neuron subtypes. We focused on determining the subtype identity of Str-projecting 5-HT neurons, since their location overlaps with the distributions of multiple 5-HT neuron subtypes. Neurons projecting to Str were labeled using axon-infecting unpseudotyped SADΔG B19 rabies viruses (B19G-SADΔG-tdTomato) injected into the ventrolateral Str. Rabies-infected neurons were identified by expression of the viral transcript *B19N* using *FISH* (RNAscope probe: RABV-gp1), and 5-HT neurons were identified based on expression of the serotonin reuptake transporter gene *Slc6a4* (*Figure 7—figure*

supplement 1). The subtype identity of RbV-infected 5-HT neurons was identified by expression of genes that were highly enriched in each subtype: *Trh* for subtype 5-HT-I, *Pdyn* for subtype 5-HT-III, *Cbln2* and *Slc17a8* for subtype 5-HT-IV. *ISH* for each of these subtype-enriched genes was performed on separate sets of RbV-labeled cells (*Figure 7*).

We found that Str-projecting DRN 5-HT neurons expressed the subtype 5-HT-III enriched gene *Pdyn* (52%, 39 of 75 RbV-labeled 5-HT neurons) or the 5-HT-IV enriched gene *Cbln2* (46%, 49 of 106 RbV-labeled 5-HT neurons), consistent with the overlap in the distribution of this projection-defined subpopulation with the inferred distributions of these two distinct subtypes. A subset of Str-projecting neurons also expressed *Slc17a8* (57%, 70 of 123 RbV-labeled 5-HT neurons), which is enriched in 5-HT-IV but also expressed in 5-HT-III at lower levels. Few Str-projecting neurons expressed *Trh* (4%, 4 of 102 RbV-labeled 5-HT neurons). This low proportion of *Trh*-expressing neurons is consistent with the low *Trh* expression in a small proportion of *Pdyn*+ 5-HT neurons (*Figure 4H*), and suggests that Str-projecting 5-HT neurons are unlikely to be subtype 5-HT-I neurons. Therefore, our results support the hypothesis that the Str-projecting DRN 5-HT neuron subpopulation is molecularly heterogeneous, and is comprised of at least 2 transcriptionally distinct 5-HT neuron subtypes.

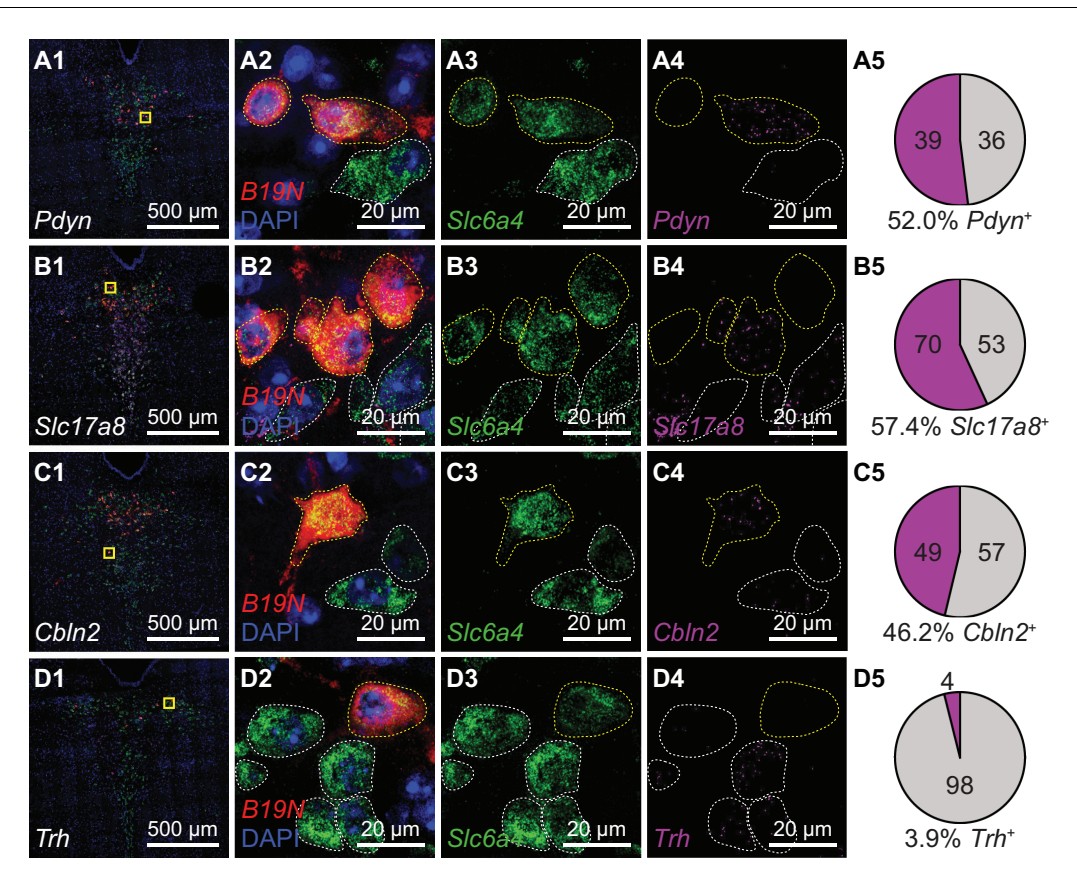

**Figure 7.** Striatum receives inputs from multiple DRN 5-HT neuron subtypes. Multiplexed *FISH* was used to identify rabies-infected Str-projecting 5-HT neurons in the DRN and quantify their expression of 5-HT subtype enriched genes. Rabies-infected neurons are identified by expression of the virus transcript *B19N*. Confocal images of the DRN from representative coronal sections are shown in column 1, and regions marked in yellow are expanded and separated by channel in columns 2–4. In columns 2–4, rabies-infected cells (*B19N*+) are outlined in yellow, whereas uninfected 5-HT neurons are outlined in white. The proportions of DRN 5-HT neurons that express each subtype-enriched gene are summarized in pie charts in column 5. Approximately half of the Str-projecting 5-HT neurons express the 5-HT-III enriched gene *Pdyn* (row A), and a similar proportion express 5-HT-IV enriched genes such as *Slc17a8* (row B) and *Cbln2* (row C). Most Str-projecting 5-HT neurons do not express the 5-HT-I enriched gene *Trh* (row D).
DOI: https://doi.org/10.7554/eLife.46464.019

The following figure supplement is available for figure 7:

**Figure supplement 1.** Quantification of RNAscope puncta in Str-projecting neurons.
DOI: https://doi.org/10.7554/eLife.46464.020

## Innervation of basal ganglia by a *Pdyn*-expressing 5-HT neuron subtype

The distribution of 5-HT-III subtype neurons overlapped with that of several projection-defined subpopulations that innervated nuclei of the basal ganglia beyond Str. To validate that the 5-HT-III subtype innervates the basal ganglia, we used an intersectional genetic targeting strategy to perform anterograde tracing from this subtype based on its enriched expression of *Pdyn*. We generated *Pet1-Flpe;Pdyn-IRES-Cre;LSL-FSF-ReaChR:mCitrine* mice in which *Pdyn*$^+$;*Pet1*$^+$ co-expressing neurons and their processes are fluorescently labeled by mCitrine-tagged ReaChR. To compare the innervation patterns of this subtype with that of the full DRN 5-HT neuron population, we also performed anterograde tracing from DRN *Pet1*$^+$ neurons by injecting Cre-dependent adeno-associated viruses (AAVs) into the DRN of *Pet1-Cre* mice, labeling *Pet1*$^+$ neurons in the dorsal and ventral DRN with EGFP and tdTomato respectively (*Figure 8A*). mCitrine-labeled axons from *Pdyn*$^+$;*Pet1*$^+$ neurons were found in a subset of regions innervated by DRN *Pet1*$^+$ neurons (*Figure 8B–F*, *Figure 8—figure supplement 1*). Of the regions innervated by both DRN *Pet1*$^+$ neurons and *Pdyn*$^+$;*Pet1*$^+$ neurons, the output nuclei of the basal ganglia, including the ventral pallidum (VP), entopeduncular nucleus (EP), and substantia nigra pars reticulata (SNr), were most densely innervated by *Pdyn*$^+$;*Pet1*$^+$ neurons (*Figure 8B–D*). In contrast, sensory structures, such as the dLGN (*Figure 8E*), and limbic structures, such as the hypothalamus (*Figure 8F*) were not innervated by *Pdyn*$^+$;*Pet1*$^+$ neurons, despite receiving dense input from DRN *Pet1*$^+$ neurons. These results therefore demonstrate that the *Pdyn*$^+$ subtype 5-HT-III neurons broadly but selectively innervate structures that are either a part of or functionally linked to the basal ganglia, and are unlikely to be involved in sensory modulation.

## Discussion

Here we present a transcriptomic map of cell types in and around the DRN, and outline the spatial organization of these cell types based on information contained in their molecular profiles. By overlaying transcriptional and anatomical features onto a common reference atlas, we describe the molecular basis for the organization of DRN 5-HT neurons into distinct subsystems. We uncovered 5 subtypes of DRN 5-HT neurons based on their gene expression profiles and showed that these are organized into distinct spatial domains within the DRN that are segregated along all three major anatomical axes. However, the spatial organization of molecularly defined 5-HT neurons does not conform to anatomical subdivisions as defined by cytoarchitecture of the DRN or the projection targets of 5-HT axons. The overlapping spatial distributions of 5-HT subtypes demonstrate the need for precise molecular and genetic approaches to dissect the function of each subtype. We also define a *Pdyn*-enriched 5-HT neuron subtype that is well-positioned to broadly but specifically alter the input and output of basal ganglia circuits, potentially mediating the modulation of motor control, decision making, and reinforcement learning by internal state.

### Anatomical organization of DRN 5-HT neuron subtypes

We present a physical map combining both transcriptional and anatomical features and show that molecularly distinct DRN 5-HT neuron subtypes are spatially segregated into distinct anatomical subdomains and innervate different but overlapping targets. We reconstructed the spatial distribution of each 5-HT subtype using spatial correlation analysis of gene differentially expressed between subtypes and showed that each cytoarchitecturally-defined DRN subregion contains different molecularly-defined 5-HT subtypes. The subregional distribution of 5-HT subtypes relates well to past studies reporting subregion-specific anatomical and molecular differences that are consistent with our integrative map of 5-HT subtype features.

Of the five subtypes, the 5-HT-V was the most transcriptionally distinct and spatially restricted, and was found in a tight dorsomedial cluster in the posterior B6 tail of the DRN. Previous reports have suggested that the B6 5-HT neuron cluster is transcriptionally more similar to 5-HT neurons in the median raphe nucleus, despite being a part of the cytoarchitecturally-defined DRN (*Commons, 2015*). This has led to the proposal that the posterior DRN is an extension of the median raphe nucleus (MRN), although this remains to be validated using single cell transcriptomic profiling of both DRN and MRN 5-HT neurons. At the molecular level, 5-HT-V subtype neurons express a distinct set of receptors compared to the other four DRN 5-HT neuron subtypes. The localization of this subtype to the dorsomedial posterior B6 tail of the DRN also suggests that this subtype may

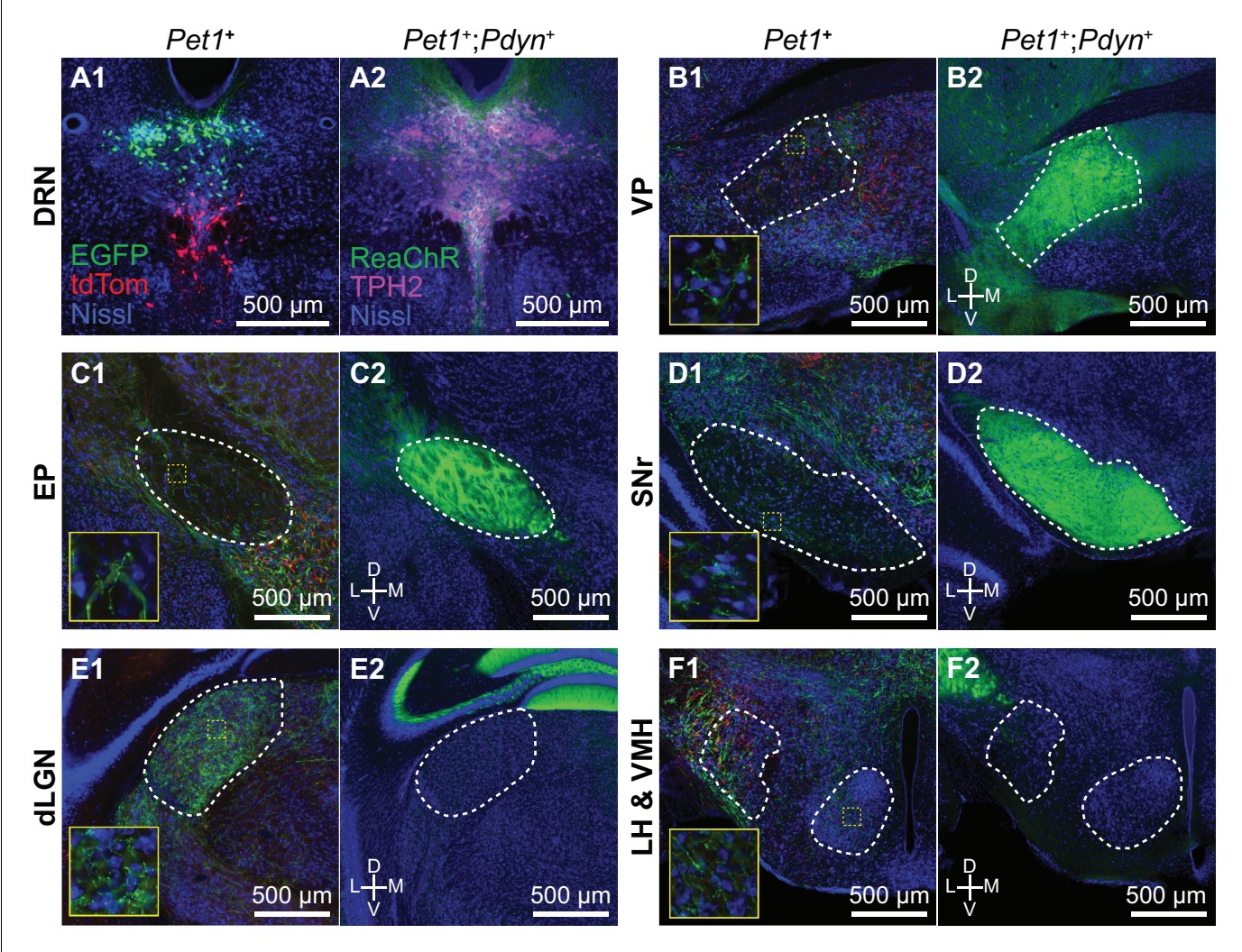

**Figure 8.** *Pdyn*[+] 5-HT neurons target basal ganglia circuits. (**A**) Conditional anterograde tracing from subsets of 5-HT neurons. Anterograde tracing from *Pet1*[+] DRN neurons was performed by injecting AAV2/1-CAG-FLEX-EGFP into the dorsal DRN and AAV2/1-CAG-FLEX-tdTomato into the ventral DRN of *ePet1-Cre* mice. A coronal section from the DRN injection site containing *Pet1*[+] cells expressing either EGFP or tdTomato is shown on the left in (A1). *Pdyn*[+];*Pet1*[+] neurons were labeled by intersectional genetic targeting with a Cre- and Flp-dependent reporter line *R26-LSL-FSF-ReaChR: mCitrine* crossed to *Pdyn-IRES-Cre;Pet1-Flpe* mice. *Pdyn*[+];*Pet1*[+] expressing ReaChR:mCitrine in the DRN are shown on the right in (A2). 5-HT neurons are labeled by immunostaining for TPH2. (**B-F**) Axonal projections of *Pdyn*[+];*Pet1*[+] neurons were compared to projections of DRN *Pet1*[+] neurons using separate approaches for conditional anterograde tracing. *Pdyn*[+];*Pet1*[+] neurons densely innervate many output nuclei of the basal ganglia, including VP (B2), EP (C2), and SNr (D2), which are also structures innervated by DRN *Pet1*[+] neurons (B1, C1, and D1). *Pdyn*[+];*Pet1*[+] axons were not found in structures that were functionally distant from basal ganglia circuits, such as dLGN (E2) and hypothalamic nuclei (F2), although these structures received dense inputs from DRN *Pet1*[+] neurons (E1 and F1). Regions outlined in yellow in B1 – F1 are enlarged in the insets showing EGFP[+]/tdTomato[+] axon fibers in each region. Abbreviations: DRN – dorsal raphe nucleus. VP – ventral pallidum. EP – entopeduncular nucleus. SNr – substantia nigra pars reticulata. dLGN – dorsal lateral geniculate nucleus. LH – lateral hypothalamus. VMH – ventromedial hypothalamus. All tissue sections are counterstained with fluorescent Neurotrace (Nissl).

DOI: https://doi.org/10.7554/eLife.46464.021

The following figure supplement is available for figure 8:

**Figure supplement 1.** *Pet1*[+];*Pdyn*[+] neurons innervate a subset of regions targeted by DRN *Pet1*[+] neurons.

DOI: https://doi.org/10.7554/eLife.46464.022

innervate ventricular and subventriclar regions and the choroid plexus and primarily target non-neuronal cells (*Muzerelle et al., 2016*). The transcriptomic profile of this subtype we present here will aid in the development of molecular and genetic approaches for targeting this subtype, and may facilitate studies on serotonergic modulation of adult neurogenesis in the subventricular zone, as

well as production and regulation of cerebrospinal fluid (CSF) by cells in the choroid plexus and ependymal lining of the ventricles (*Banasr et al., 2004*; *Fisone et al., 1995*; *Moskowitz et al., 1979*; *Young et al., 2011*).

The ventral DRN, comprised of the ventromedial subregion (or DRV) and interfascicular subregion (or DRI), contains mostly *Slc17a8*-enriched 5-HT-IV subtype neurons that innervate cortex (*Ren et al., 2018*). We noted that 5-HT-IV neurons were distributed throughout most of the anterior-posterior extent of the ventral DRN, although it is suggested that there is a topographical organization of cortical projecting 5-HT neurons along this axis. We speculate that this may reflect the broad and divergent cortical projections of individual 5-HT-IV neurons that collectively span most of the cerebral cortex. Alternatively, additional heterogeneity within the 5-HT-IV subtype may explain variation along the anterior-posterior axis that may be revealed with deeper sequencing.

In comparison, the dorsolateral DRN, also called the lateral wings of the DRN (lwDRN) and ventrolateral DRN (DRVL), showed strongly negative correlations with the 5-HT-IV subtype, particularly in the posterior half of the DRN where there is clearer spatial separation of $Tph2^+$ neurons into medial and lateral compartments. Subtypes 5-HT-I and 5-HT-II were intermingled in this region, which is enriched in *Trh*, a 5-HT-I enriched gene. The dorsolateral subregion innervates many subcortical sensory and hypothalamic nuclei, in addition to motor nuclei in the brainstem (*Lee et al., 2008*; *Muzerelle et al., 2016*). We speculate that 5-HT-I and 5-HT-II form the cellular bases for the separation of DRN 5-HT projections to motor/limbic regions versus sensory regions respectively, given the expression of the TRH receptor gene (*Trhr*) in limbic regions, such as the NAc shell, the bed nucleus of the stria terminalis, and hypothalamus, as well as several brainstem motor nuclei innervated by the DRN, such as the facial motor nucleus (Allen Brain Atlas). Further experiments using retrograde tracing and molecular profiling of DRN 5-HT neurons innervating these structures can be performed to evaluate these hypotheses. Additionally, the close spatial proximity of these two subtypes also presents a plausible mechanism for competitive inhibition between these separate subsystems via 5-HT1A receptors and local 5-HT release, which may be relevant to selective sensory modulation by movement or changes in behavioral state (*Hall et al., 2010*; *Yokogawa et al., 2012*).

Although we show that 5-HT subtypes have distinct spatial locations, we also found considerable overlaps in their distributions, particularly within the dorosmedial subregion in the anterior DRN that contains many basal ganglia-projecting 5-HT neurons. While the *Pdyn*-expressing 5-HT-III subtype was localized to the dorsomedial subregion, subtypes 5-HT-I and 5-HT-II were also found in this subregion in the anterior third of the DRN despite being distributed to dorsolateral subregions in more posterior sections. Furthermore, a detailed comparison of the 5-HT subtype distributions with cytoarchitecturally-defined subregion boundaries showed that subtype distributions are not confined within these subregion boundaries, especially in the anterior half of the DRN. The poor correspondence between molecularly-inferred and cytoarchitecturally-defined spatial domains cautions against the use of anatomical subregions for targeting 5-HT subsystems.

## Molecular organization of DRN 5-HT subsystems

5-HT neurons are organized into discrete subsystems that (i) have distinct and dissociable behavioral functions, and (ii) are comprised of 5-HT neurons possessing unique molecular, physiological, and anatomical properties. However, the complex architecture of the DRN has impeded efforts to understand the interrelationship of these three aspects of 5-HT diversity, as necessary to study the differential properties and functions of each subsystem and its distinct subtype of 5-HT neuron. Given the extensive overlaps in the distribution of different 5-HT neuron subsets at the anatomical level, we were motivated by the hypothesis that a transcriptome-based molecular definition of 5-HT neuron subtypes was required to both understand the organization of heterogeneous 5-HT neuron subtypes as well as to enable their precise targeting for functional analyses. Consistent with this hypothesis, our results suggest that many of the physiological and anatomical differences between 5-HT neuron subtypes are a result of molecular differences specified by expression of overlapping but distinct transcriptional networks.

Some key technical limitations of scRNA-seq, particularly for droplet-based methods such as the inDrop platform used in this study, are the relatively low transcript capture efficiency and low number of reads per cell. The higher occurrence of drop-out events, which are failures in detecting expressed transcripts in a cell, also makes it more difficult to detect low copy transcripts that may be important for distinguishing between highly similar subtypes, such as the 5-HT-I and 5-HT-II subtypes

in our study. Thus our study does not seek to present a definitive catalogue of all neuron subtypes present in the DRN, since the techniques used are limited in their ability to resolve finer levels of heterogeneity and differential expression of low copy transcripts. In spite of these limitations, our study still successfully captures a substantial amount of transcriptional heterogeneity to resolve differences between neuron subtypes that are informative of their diverse morphological and functional features. Furthermore, we find differential expression of transcription factors across subtypes that may underlie many of these molecular differences that are specified during the development of each subtype. The dynamics of these gene expression programs can be further elucidated by future studies that sample multiple time points along the developmental trajectory of the DRN. Further studies incurring a trade-off on cell count and the number of cell types captured for higher sequencing depth may also resolve additional transcriptional heterogeneity between subtypes within specific populations. Eventually these molecular approaches will be well complemented by whole-brain reconstructions of axons of single neurons (*Economo et al., 2016*; *Gong et al., 2016*) as well as spatial transcriptomics (*Chen et al., 2015*; *Wang et al., 2018*).

## Local interactions between DRN cell types

Our scRNA-seq dataset yields predictions about local interactions involving DRN cells, including those with non-neuronal cells, based the expression patterns of known receptor-ligand pairs. For example, the expression of *Hdc* in ependymal cells suggests that ependymal cells may signal to nearby neurons via non-synaptic release of histamine. Histaminergic paracrine signaling from ependymal cells can exert differential effects on various neuronal subtypes based on proximity to the cerebral aqueduct and differential expression of histaminergic receptors. Notably, the histamine receptors *Hrh1* and *Hrh3* are expressed in 5-HT subtypes 5-HT-I, 5-HT-II, and 5-HT-III, which are all located in dorsal DRN subregions in close proximity to the cerebral aqueduct. In contrast, 5-HT-IV subtype neurons, which are in ventral subregions further from the aqueduct, do not express detectable levels of *Hrh1* and express *Hrh3* at very low levels. Enrichment of the $G_i$-coupled *Hrh3* receptor in 5-HT-III further suggests that histamine may inhibit this subtype while increasing the excitability of 5-HT-I and 5-HT-II neurons. Release of peptides such as *Pdyn* from a subset of ependymal cells can also have differential effects dependent on the expression pattern of the κ-opioid receptor. Additionally, ependymal cells may communicate via secretion of these transmitters and peptides into the CSF. Transmitters and peptides released into the ventricular system can have long range effects facilitated by the flow of CSF and signal to neuronal cells extending processes to the ventricular walls, including some 5-HT neurons, as well as non-neuronal cells including ventricular immune cells and choroid plexus cells.

We also found that pvMΦs within or in close proximity to the DRN express the $G_i$-coupled *Htr1b* receptor, unlike pvMΦs in brain regions that do not contain 5-HT neurons. This suggests that pvMΦs in different brain regions may be specialized via differential expression of neurotransmitter receptors to be specifically tuned to monitor their local environment and respond to changes in neuronal activity. Activation of $G_i$-coupled pathways on pvMΦs via 5-HT1B receptors is also a potential mechanism for the regulation of macrophage polarization and inflammatory responses by tonic 5-HT release within the DRN, which may be relevant to the increasing importance of neuroinflammatory pathways in the etiology of psychiatric disorders (*Brisch et al., 2017*; *Howerton et al., 2014*; *Watkins et al., 2014*).

The diversity of neuronal subtypes in and around the DRN also highlights the importance of elucidating local circuitry with knowledge of the molecular identity of both pre- and post-synaptic neurons. For direct signaling between 5-HT neurons, somatodendritic release of 5-HT and activation of inhibitory 5-HT1A receptors on adjacent 5-HT neurons is likely to mediate lateral inhibition between the 5-HT neurons of the same subtype, due to the spatial clustering of the subtypes. However, GABAergic and/or glutamatergic neurons may be excited by adjacent 5-HT neurons via activation of their 5-HT2C and 5-HT7 receptors to mediate indirect interactions between 5-HT neuron subtypes. These interactions may be either synergistic if mediated by glutamatergic neurons innervating other 5-HT subtypes, or antagonistic in a form of lateral inhibition via GABAergic interneurons. Subtype-specific connections between 5-HT neurons and local GABAergic and glutamatergic neurons can mediate interactions between different subsystems and may contribute to the contrasting and opposing functions ascribed to the DRN as a whole. Although 5-HT neurons have been reported to form local connections (*Weissbourd et al., 2014*), it is currently unknown how the 18 neuronal

subtypes we have described are organized into local circuits. To generate models of these local circuits, the connectivity between DRN and ventrolateral PAG neurons will need to be mapped with subtype-level molecular resolution. This will likely require the use of retrograde tracing techniques with improved specificity for targeting neuronal subtypes in combination with spatial transcriptomic methods that are compatible with the large volume of the DRN. These methods can also be extended to examine the interactions between larger networks of 5-HT subsystems distributed across the raphe nuclei, which have been shown to be interconnected based on conditional anterograde tracing using intersectional targeting of 5-HT neurons originating from different embryonic rhombomeres (*Bang et al., 2012*).

## A 5-HT subsystem targeting basal ganglia circuits

The transcriptomic information we obtained on DRN 5-HT neuron subtypes allowed us to access a specific 5-HT subsystem that innervates circuits of the basal ganglia. We showed that the subsystem comprised of the *Pdyn*-enriched subtype 5-HT-III sends broad projections to both input and output nuclei of the basal ganglia, while avoiding other sensory and limbic structures, which have distinct functions from the basal ganglia and are likely to receive inputs from other DRN 5-HT neuron subtypes. Although we also showed that the basal ganglia receives inputs from multiple 5-HT neuron subtypes, particularly in the case of the Str-projecting subpopulation, the dense innervation of basal ganglia output nuclei by 5-HT-III neurons suggests that this 5-HT subsystem has a prominent role in the regulation of motor output and action selection through its effects on the activity of basal ganglia output nuclei. We also speculate that 5-HT release from this subsystem may have different effects on distinct output channels of the basal ganglia, given the differential expression of 5-HT receptors on distinct subpopulations of SNr and EP projection neurons (*Saunders et al., 2018*; *Wallace et al., 2017*). The targeting of distinct channels or subcircuits within the basal ganglia may also be consistent with the innervation of the striatum by multiple 5-HT neuron subtypes given the topographical arrangement of convergent cortical inputs into distinct domains within the striatum (*Hintiryan et al., 2016*), since it is plausible that distinct striatal domains are each innervated by a different but functionally related 5-HT neuron subtype. Subcircuit-specific effects of 5-HT within the basal ganglia would also have implications on models for the function of 5-HT in decision-making and motor control (*Miguelez et al., 2014*; *Reed et al., 2013*), although further work is needed to elucidate the structural and functional organization of these subcircuits in the basal ganglia (*Calabresi et al., 2014*).

Our results provide a resource for the design of intersectional strategies for the functional dissection of DRN cell types. The *LSL-FSF-ReaChR:mCitrine* intersectional reporter line that we used for conditional anterograde tracing will also be suitable for optogenetic perturbation experiments in vivo to investigate the circuit and behavioral functions of these distinct 5-HT subsystems. Greater spatial specificity can be achieved through the use of tapered optic fibers for site-specific light delivery (*Pisanello et al., 2014*), intersectional viral constructs (*Fenno et al., 2014*) optimized for expression in cell types of interest, and the development of additional Cre and Flp recombinase driver lines with improved specificity to increase the number of subtypes and subsystems that can be accessed through genetic and molecular approaches. The functional studies facilitated by the development of these additional tools will provide further mechanistic insights into the coordinated action of distinct cell types and subsystems of the DRN in the regulation of behavior.

## Materials and methods

**Key resources table**

| Reagent type (species) or resource | Designation | Source or reference | Identifiers | Additional information |
|---|---|---|---|---|
| Strain (*Mus musculus*) | C57BL/6J | The Jackson Laboratory | Cat# JAX:000664, RRID:IMSR_JAX:000664 | - |
| Genetic reagent (*Mus musculus*) | *Pet1-Cre* | The Jackson Laboratory | Cat# JAX:012712, RRID:IMSR_JAX:012712 | - |

*Continued on next page*

*Continued*

| Reagent type (species) or resource | Designation | Source or reference | Identifiers | Additional information |
|---|---|---|---|---|
| Genetic reagent (*Mus musculus*) | *Pet1-Flpe* | PMID: 18344997 | RRID:MGI:5004974 | Lab of Susan Dymecki |
| Genetic reagent (*Mus musculus*) | *Pdyn-IRES-Cre* | The Jackson Laboratory | Cat# JAX:027958, RRID:IMSR_JAX:027958 | Lab of Bradford Lowell |
| Genetic reagent (*Mus musculus*) | *R26-LSL-FSF-ReaChR:mCitrine* | The Jackson Laboratory | Cat# JAX:024846, RRID:IMSR_JAX:024846 | - |
| Genetic reagent (Adeno-associated virus) | AAV2/1-CAG-FLEX-EGFP | Penn Vector Core | RRID:Addgene_51502 | $10^{12}$ GC/ml |
| Genetic reagent (Adeno-associated virus) | AAV2/1-CAG-FLEX-tdTomato | Penn Vector Core | RRID:Addgene_51503 | $10^{12}$ GC/ml |
| Genetic reagent (G-deleted rabies virus) | B19G-SADΔG-EGFP; RbV-EGFP | Other | N/A | Generated in-house (see Materials and methods) $10^9$ IU/ml |
| Genetic reagent (G-deleted rabies virus) | B19G-SADΔG-tdTomato; RbV-tdTomato | Other | N/A | Generated in-house (see Materials and methods) $10^9$ IU/ml |
| Antibody | Anti-TPH2 (rabbit polyclonal) | Millipore | Cat# ABN60, RRID:AB_10806898 | 1:1000 |
| Antibody | Anti-rabbit A647 (goat polyclonal) | ThermoFisher Scientific | Cat# A-21244, RRID:AB_2535812 | 1:500 |
| Chemical compound, drug | (R)-CPP | Tocris Bioscience | Cat# 0247 | 10 µM |
| Chemical compound, drug | Tetrodotoxin | Tocris Bioscience | Cat# 1069 | 1 µM |
| Chemical compound, drug | Kynurenic acid | Millipore Sigma | Cat# K3375 | 0.9 mM |
| Chemical compound, drug | Triptolide | Millipore Sigma | Cat# T3652 | 10 µM |
| Chemical compound, drug | Actinomycin D | Millipore Sigma | Cat# A9415 | 5 µg/ml |
| Chemical compound, drug | Anisomycin | Millipore Sigma | Cat# A5862 | 30 µg/ml |
| Peptide, recombinant protein | Pronase | Millipore Sigma | Cat# 10165921001 | 1 mg/ml |
| Commercial assay, kit | Papain Dissociation System, without EBSS | Worthington | Cat# LK003163 | 20 U/ml papain 0.05 mg/ml DNAse I |
| Commercial assay, kit | RNAscope V1 fluorescent multiplex detection assay reagents | ACDBio | Cat# 320851 | - |
| Commercial assay, kit | RNAscope V1 fluorescent multiplex detection assay probes | ACDBio | Cat# 428551; Cat# 498971; Cat# 400951; Cat# 318771; Cat# 525191; Cat# 319171; Cat# 431261; Cat# 319191; Cat# 315851; Cat# 436811; Cat# 525201; Cat# 456781 | *Cbln2* *Crhbp* *Gad1* *Pdyn* *Prkcq* *Slc17a6* *Slc17a8* *Slc32a1* *Slc6a4* *Trh* *Trpc3* *V-RABV-gp1* |

*Continued on next page*

*Continued*

| Reagent type (species) or resource | Designation | Source or reference | Identifiers | Additional information |
|---|---|---|---|---|
| Software, algorithm | inDrops pipeline (Python) | PMID: 26000487 | N/A | https://github.com/indrops/indrops |
| Software, algorithm | R 3.4.4 | R Project for Statistical Computing | RRID:SCR_001905 | https://cran.r-project.org/ |
| Software, algorithm | Seurat 2.3.1 | PMID: 29608179 | RRID:SCR_016341 | https://satijalab.org/seurat/ |
| Software, algorithm | MAST 1.4.1 | PMID: 26653891 | RRID:SCR_016340 | https://bioconductor.org/packages/release/bioc/html/MAST.html |
| Software, algorithm | goseq 1.30.0 | PMID: 20132535 | RRID:SCR_017052 | https://bioconductor.org/packages/release/bioc/html/goseq.html |
| Software, algorithm | MATLAB (R2016b, R2018b) | MathWorks | RRID:SCR_001622 | - |
| Software, algorithm | Fiji | PMID: 22743772 | RRID:SCR_002285 | https://imagej.net/Fiji |
| Software, algorithm | Ilastik 1.2.2 | DOI: 10.1109/ISBI.2011.5872394 | RRID:SCR_015246 | https://www.ilastik.org/index.html |
| Software, algorithm | CellProfiler 3.1.5 | PMID: 17269487 | RRID:SCR_007358 | https://cellprofiler.org/ |
| Other | Neurotrace 435 | ThermoFisher Scientific | Cat# N21479 | 1:100 |
| Other | VectaShield mounting media | Vector Labs | Cat# H-1000, RRID:AB_2336789 | - |
| Other | ProLong Gold mounting media | ThermoFisher Scientific | Cat# P36934 | - |

## Mice

The following mouse strains/lines were used in this study: C57BL/6J (The Jackson Laboratory, Stock # 000664), *Pet1-Cre* (The Jackson Laboratory, Stock # 012712), *Pet1-Flpe* (from the lab of Susan Dymecki [*Jensen et al., 2008*]), *Pdyn-IRES-Cre* (The Jackson Laboratory, Stock # 027958), and *R26-LSL-FSF-ReaChR:mCitrine* (The Jackson Laboratory, Stock # 024846). Animals were kept on a 12:12 regular light/dark cycle under standard housing conditions. All procedures were performed in accordance with protocols approved by the Harvard Standing Committee on Animal Care following guidelines described in the U.S. National Institutes of Health Guide for the Care and Use of Laboratory Animals.

## Adeno-Associated viruses (AAVs)

Recombinant AAVs used for anterograde tracing experiments (AAV2/1-CAG-FLEX-EGFP, AAV2/1-CAG-FLEX-tdTomato) were commercially obtained from the Penn Vector Core. Virus aliquots were stored at −80°C, and were injected at a concentration of approximately $10^{12}$ GC/ml.

## Rabies viruses

Rabies viruses used for retrograde tracing (B19G-SADΔG-EGFP, B19G-SADΔG-tdTomato) were generated in-house (*Osakada and Callaway, 2013*; *Wickersham et al., 2010*). Virions were amplified from existing stocks in several rounds of low-MOI passaging through BHK-B19G cells by transferring filtered supernatant, with 3 to 4 days between passages a maximum of 3 passages. Cells were grown at 35°C and 5% $CO_2$ in DMEM with GlutaMAX (Thermo Scientific, #10569010) supplemented with 5% heat-inactivated FBS (Thermo Scientific #10082147) and antibiotic-antimycotic (Thermo Scientific #15240–062). Media containing virions were collected at the end of the last passaging round and incubated with benzonase nuclease (1:1000, Millipore #70664) at 37°C for 30 min, followed by filtration through a 0.22 μm PES filter. Filtered supernatant was transferred to ultracentrifuge tubes (Beckman Coulter #344058) with 2 ml of a 20% sucrose in dPBS cushion and ultracentrifugated at 20,000 RPM (Beckman Coulter SW 32 Ti rotor) at 4°C for 2 hr. The supernatant was discarded and

the pellet was resuspended in dPBS for 6 hr on an orbital shaker at 4°C before aliquots were prepared and frozen for long-term storage at −80°C. Unpseudotyped rabies virus titers were estimated using a serial dilution method counting infected HEK 293 T cells and quantified as infectious units per ml (IU/ml).

## Stereotaxic surgeries

Mice were initially anesthetized with 5% isoflurane (80% oxygen) and maintained at 1–2.5% isoflurane after placement on the stereotaxic frame (David Kopf Instruments, Model 1900 Stereotaxic Alignment System). The scalp was cleaned and sterilized before an incision was made to expose the skull, and sterile ophthalmic ointment was applied to the eyes. For leveling the horizontal plane, a stereotaxic alignment tool (David Kopf Instruments, Model 1905) was used to zero the relative dorsoventral displacement of Bregma and Lambda, as defined in the Paxinos and Franklin Mouse Brain Altas (*Paxinos and Franklin, 2001*), for adjusting tilt of the anterior-posterior axis, and of two points equidistant to the left and right of Bregma for adjusting the tilt of the medial-lateral axis. Craniotomies were prepared using a mounted drill (David Kopf Instruments, Model 1911) with careful removal of the bone flap and overlying dura using forceps and a fine needle tip, and were covered with sterile 0.9% saline before and during the injection to prevent desiccation. Viruses were front-filled into a pulled glass pipette (Drummond Scientific, #5-000-2005) filled with mineral oil (Millipore Sigma, M3516) and connected to a 5 µl Hamilton syringe (Hamilton #84850) via polyethylene tubing filled with mineral oil. Glass pipettes were pulled to obtain a tip size of approximately 40–60 µm on a pipette puller (Sutter Instrument Co., P-97). Viruses were infused into target regions at approximately 100 nl/min using a syringe pump (Harvard Apparatus, #883015), and pipettes were slowly withdrawn (<10 µm/s) at least 10 min after the end of the infusion. Following wound closure, mice were placed in a cage with a heating pad until their activity was recovered before returning to their home cage. Mice were given pre- and post-operative oral carprofen (MediGel CPF, 5 mg/kg/day) as an analgesic, and monitored daily for at least 4 days post-surgery.

## Stereotaxic injection coordinates and volumes

All coordinates are relative to Bregma along the anterior-posterior axis and medial-lateral axis, and relative to the pial surface along the dorsoventral axis. 'BL' denotes the distance between Bregma and Lambda (as defined in the Paxinos mouse brain atlas). Angles for DV coordinates are relative to the dorsoventral axis along the sagittal plane, with 0° being a straight vertical approach and negative values denoting a posterior-to-anterior approach. Dorsal raphe nucleus (DRN): AP = -(6.25 * BL/4.35) mm, ML = 0.00 mm, DV = −3.33 mm at −40°, 200–300 nl. Ventrolateral striatum (vlStr): AP = +0.40 mm, ML = ±2.45 mm, DV = −3.10 mm at 0°, 300 nl. Dorsal lateral geniculate nucleus (dLGN): AP = -(2.00 * BL/4.20) mm, ML = ±2.25 mm, DV = −3.00 mm at 0°, 150 nl. Globus pallidus, external (GPe): AP = +0.14 mm, ML = ±2.34 mm, DV = −3.90 mm at 0°, 200 nl. Lateral hypothalamic area (LHA): AP = −1.46 mm, ML = ±1.10 mm, DV = −5.10 mm at 0°, 150 nl. Habenula (Hb): AP = -(1.30 * BL/4.2) mm, ML = ±0.43 mm, DV = −2.80 mm at 0°, 50 nl. Nucleus accumbens core (NAcc): AP = +1.54 mm, ML = ±1.10 mm, DV = −4.00 mm at 0°, 300 nl. Primary motor cortex (M1), six sites: AP = {+2.10 mm, +1.60 mm, +1.10 mm}, ML = {±1.75 mm, ±1.60 mm,±1.60 mm}, DV = {[−1.20 mm, −0.60 mm], [−1.10 mm, −0.50 mm], [−1.00 mm, −0.40 mm]} at 0°, 150 nl per site. Medial prefrontal cortex (mPFC), four sites: AP = {+2.4 mm, +1.8 mm}, ML = ±0.40 mm, DV = {−1.65 mm, −1.25 mm} at 0°, 150 nl per site.

## Single cell dissociation and RNA sequencing

8-to 10 week old C57BL/6J mice were pair-housed in a regular 12:12 light/dark cycle room prior to tissue collection. Mice were transcardially perfused with an ice-cold choline cutting solution containing neuronal activity blockers (110 mM choline chloride, 25 mM sodium bicarbonate, 12 mM D-glucose, 11.6 mM sodium L-ascorbate, 10 mM HEPES, 7.5 mM magnesium chloride, 3.1 mM sodium pyruvate, 2.5 mM potassium chloride, 1.25 mM sodium phosphate monobasic, 10 µM (R)-CPP, 1 µM tetrodotoxin, saturated with bubbling 95% oxygen/5% carbon dioxide, pH adjusted to 7.4 using sodium hydroxide). Brains were rapidly dissected out and sliced into 250 µm thick coronal sections on a vibratome (Leica Biosystems, VT1000) in a chilled cutting chamber filled with choline cutting solution. Coronal slices containing the DRN were then transferred to a chilled dissection dish

containing a choline-based cutting solution for microdissection. Fiber tracts were used for guiding the microdissection but were excluded from the collected tissue chunk. Dissected tissue chunks were transferred to cold HBSS-based dissociation media (Thermo Fisher Scientific Cat. # 14170112, supplemented to final content concentrations: 138 mM sodium chloride, 11 mM D-glucose, 10 mM HEPES, 5.33 mM potassium chloride, 4.17 mM sodium bicarbonate, 2.12 mM magnesium chloride, 0.9 mM kynurenic acid, 0.441 mM potassium phosphate monobasic, 0.338 mM sodium phosphate monobasic, 10 µM (R)-CPP, 1 µM tetrodotoxin, saturated with bubbling 95% oxygen/5% carbon dioxide, pH adjusted to 7.35 using sodium hydroxide) supplemented with an additional inhibitor cocktail (10 µM triptolide, 5 µg/ml actinomycin D, 30 µg/ml anisomycin) and kept on ice until dissections were completed. Dissected tissue chunks for each sample were pooled into a single tube for the subsequent dissociation steps. Tissue chunks were first mixed with a digestion cocktail (dissociation media, supplemented to working concentrations: 20 U/ml papain, 1 mg/ml pronase, 0.05 mg/mL DNAse I, 10 µM triptolide, 5 µg/ml actinomycin D, 30 µg/ml anisomycin) and incubated at 34°C for 90 min with gentle rocking. The digestion was quenched by adding dissociation media supplemented with 0.2% BSA and 10 mg/ml ovomucoid inhibitor (Worthington Cat. # LK003128), and samples were kept chilled for the rest of the dissociation procedure. Digested tissue was collected by brief centrifugation (5 min, 300 *g*), re-suspended in dissociation media supplemented with 0.2% BSA, 1 mg/ml ovomucoid inhibitor, and 0.05 mg/mL DNAse I. Tissue chunks were then mechanically triturated using fine-tip plastic micropipette tips of progressively decreasing size. The triturated cell suspension was filtered in two stages using a 70 µm cell strainer (Miltenyi Biotec Cat # 130-098-462) and 40 µm pipette tip filter (Bel-Art Cat. # H136800040) and washed in two repeated centrifugation (5 min, 300 *g*) and re-suspension steps to remove debris before a final re-suspension in dissociation media containing 0.04% BSA and 15% OptiPrep (Sigma D1556). Cell density was calculated based on hemocytometer counts and adjusted to approximately 100,000 cells/ml. Single-cell encapsulation and RNA capture on the InDrop platform was performed at the Harvard Medical School ICCB Single Cell Core using v3 chemistry hydrogels based on previously described protocols (*Zilionis et al., 2017*). Suspensions were kept chilled until the cells were flowed into the microfluidic device. Libraries were prepared and indexed following the protocols referenced above, and sequencing-ready libraries were stored at −80°C. Libraries from different samples were pooled and sequenced on an Illumina NextSeq 500 (High Output v2 kits).

## Sequencing data processing

NGS data were processed using a previously published pipeline in Python available at https://github.com/indrops/indrops (*Klein et al., 2015*). Briefly, reads were filtered by expected structure and sorted by the corresponding library index. Valid reads were then demultiplexed and sorted by cell barcodes. Cell barcodes containing fewer than 250 total reads were discarded, and remaining reads were aligned to a reference mouse transcriptome (Ensembl GRCm38 release 87) using Bowtie 1.2.2 (m = 200, n = 1, l = 15, e = 100). For alignment, the mouse transcriptome was modified with the addition of genes from the SAD B19 rabies viruses and transgenes (*B19N, B19P, B19M, B19L, EGFP, tdTomato, AmCyan1*). Aligned reads were then quantified as UMI-filtered mapped read (UMIFM) counts. UMIFM counts and quantification metrics for each cell were combined into a single file sorted by library and exported as a gunzipped TSV file.

## Pre-Clustering filtering and normalization

Analysis of the processed NGS data was performed in R (version 3.4.4) using the *Seurat* package (version 2.3.1) (*Butler et al., 2018*; *Satija et al., 2015*). A custom R script was used to combine the expression data and metadata from all libraries corresponding to a single batch, and cells with fewer than 500 UMIFM counts were removed. The expression data matrix (Genes x Cells) was filtered to retain genes with >5 UMIFM counts, and then loaded into a *Seurat* object along with the library metadata for downstream processing. The percentage of mitochondrial transcripts for each cell (*percent.mito*) was calculated and added as metadata to the *Seurat* object. Cells in the object were further filtered using the following parameters: *Reads – min. 20000, max. Inf; nUMI – min. 500, max. 18000; nGene – min. 200, max. 6000; percent.mito – min. -Inf, max. 0.1.* Low quality libraries identified as outliers on scatter plots of quality control metrics (e.g. unusually low gradient on the *nGene* vs. *nUMI*) were also removed from the dataset. Filtered *Seurat* objects were then log-normalized at

10,000 transcripts per cell. Effects of latent variables (*nUMI, percent.mito, Sex, orig.ident*) were estimated and regressed out using a GLM (*ScaleData* function, *model.use = 'linear'*), and the scaled and centered residuals were used for dimensionality reduction and clustering.

## Dimensionality reduction and batch effect correction

Canonical correlation analysis (CCA) was used for dimensionality reduction and mitigation of batch effects. We used the union of the top 2000 highly variable genes from each batch dataset (3341 genes) and used the *RunCCA* function in Seurat to calculate 50 canonical variates (CVs). After inspection of the CVs, the first 35 CVs were used for subspace alignment when merging datasets from different batches into a single *Seurat* object. Cells with expression profiles not well explained by CVs (*var.ratio.pca* <0.5) were removed from the dataset before clustering.

## Cell clustering and cluster identification

Initial clustering was performed on the merged and CCA-aligned dataset using the first 35 CVs, and t-SNE was used only for data visualization. Clustering was run using the SNN-based *FindClusters* function using the SLM algorithm and 10 iterations. Clustering was performed at varying resolution values, and we chose a final value of 2 for the resolution parameter for this stage of clustering. Clusters were assigned preliminary identities based on expression of combinations of known marker genes for major cell classes and types. A list genes used for identifying major cell classes and types is provided in *Supplementary file 1*. Low quality cells were identified based on a combination of low gene/UMIFM counts and high levels of mitochondrial and nuclear transcripts (e.g. *Malat1*, *Meg3*, *Kcnq1ot1*) typically clustered together and were removed. Following assignment of preliminary identities, cells were divided into data subsets as separate Seurat objects (neurons, astrocytes, ependymal cells, vascular cells, immune cells, oligodendrocytes and polydendrocytes) for further subclustering. The expression matrix for each data subset was further filtered to include only genes expressed by the cells in the subset (minimum cell threshold of 0.5% of cells in the subset).

Subclustering was performed iteratively on each data subset to resolve additional cell types and subtypes. Briefly, clustering was run at high resolution, and the resulting clusters were ordered in a cluster dendrogram built using the *Ward2* method in *hcliust* using cluster-averaged gene expression for calculating the Euclean distance matrix. Putative doublets/multiplets were identified based on co-expression of known marker genes for different cell types (e.g. neuronal and glial markers, or endothelial cell and pericyte markers). Putative doublets tended to separate from other cells and cluster together, and these clusters were removed from the dataset. Cluster separation was evaluated using the *AssessNodes* function and inspection of differentially expressed genes at each node. Clusters with poor separation, based on high OOBE scores and differential expression of mostly housekeeping genes, were merged to avoid over-separation of the data. The dendrogram was reconstructed after merging or removal of clusters, and the process of inspecting and merging or removing clusters was repeated until all resulting clusters could be distinguished based on a set of differentially expressed genes that we could validate separately. For small clusters that were previously identified in preliminary analysis of the separate batches, such as the *Met*-expressing 5-HT neuron subtype, or expected clusters based on existing literature, such as the $Slc17a8^+;Fev^-$ neurons, manual identification and clustering was performed if these cells were not separated by the clustering algorithm.

## Differential expression tests

Tests for differential gene expression were performed using *MAST* version 1.4.1 (*Finak et al., 2015*) through the `FindMarkersNode` function in *Seurat* (`logfc.threshold = 0.25`, `min.pct = 0.1`). P values were corrected using the Benjamini-Hochberg method and filtered a 5% false discovery rate (Q < 0.05).

## Gene Ontology term enrichment tests

GO term enrichment tests were performed using the `goseq` package version 1.30.0 in R (*Young et al., 2010*). Genes differentially expressed between 5-HT neuron subtypes were used as the target list, and all genes expressed in the 5-HT neuron dataset were used for the background

list. Enrichment scores were calculated using the Wallenius approximation method in `goseq`, and terms were filtered using Q values at a 5% false discovery rate (Benjamini-Hochberg correction).

## Spatial correlation analysis

Spatial correlation analysis was performed in MATLAB (MathWorks, R2016b and R2018b). A spatial expression matrix for the differentially expressed (DE) genes was constructed using in situ hybridization images from the Allen Brain Atlas. Only DE genes with coronal imaging data were included in the matrix to achieve sufficient resolution along the dorsal-ventral and medial-lateral axes. For each gene, *ISH* images spanning the DRN were downsampled by a factor of 2 before registration and expression quantification using a custom MATLAB script. The midline of each slice was vertically aligned and centered, and images were and cropped to retain a region spanning approximately 1 mm across the midline containing the DRN. Cropped images were each assigned to a position along the anterior-posterior axis corresponding to a coronal section from the Allen Brain Atlas averaged coronal template (100 μm intervals). Expression was binarized by thresholding the image, and positive cells were detected using a Laplacian-of-Gaussian spot detection algorithm. Spatial expression patterns were obtained from two-dimensional histograms (bin size = 50 μm x 50 μm) of positive cells counts in each slice. Fine alignment of each 2D histogram along the dorsal-ventral axis was adjusted manually using *Tph2 ISH* images as the reference. Anatomical landmarks visible in the Nissl or background stain, such as the cerebral aqueduct and fiber tracts, were used for the fine adjustment. Aligned histograms were cropped to retain a square region (36 bins x 36 bins) containing the DRN. Cropped 2D histograms were averaged with the left-right reflection, and the *ISH* count data for each experiment (36 bins x 36 bins x 12 sections) was unrolled into a row vector. Row vectors for the same gene were averaged, ignoring NaN values from missing *ISH* data, to obtain a single row vector for each gene. Row vectors for each gene were concatenated to generate the spatial expression matrix. The spatial expression matrix (Genes x Bins) was further filtered in two stages: bins with no *Tph2* expression were removed, and genes were filtered based on total expression and mean-normalized variance (*Figure 5—figure supplement 1*). Pearson correlation coefficients were calculated between the averaged gene expression of each 5-HT neuron subtype cluster from the scRNA-seq dataset and each spatial bin in the filtered spatial expression matrix.

## Histology for anatomical tracing

Mice were deeply anesthetized with isoflurane and transcardially perfused with 5–10 ml chilled 0.1 M PBS, followed by 10–15 ml chilled 4% paraformaldehyde in 0.1 M PBS. Brains were dissected out and post-fixed overnight at 4˚C, followed by incubation in a storing/cryoprotectant solution of 30% sucrose and 0.05% sodium azide in 0.1 M PBS for at least 1–2 days to equilibrate. 50 μm coronal slices were prepared on a freezing microtome (Leica Biosystems, SM2010 R). Slices were either immunostained (see section on Immunohistochemistry), or counterstained using Neurotrace 435 (ThermoFisher Scientific N21479) at a concentration of 1:100 in 0.1 M PBS with 0.5% Triton X-100. Slices were mounted in VectaShield mounting media (Vector Labs, H-1000) and imaged on an Olympus VS120 slide scanning microscope. Confocal images were taken on an Olympus FV1000 or FV1200 confocal microscope for assessing co-localization.

## Immunohistochemistry

50 μm thick free-floating tissue sections were rinsed 3 × 5 min with 0.1 M PBS containing 0.5% Triton X-100 (PBST) and blocked in a PBST-based solution containing 6% normal goat serum for 1 hr at room temperature. The blocking solution was then removed, and slices were incubated overnight at 4˚C in a PBST-based antibody carrier solution containing 3% normal goat serum and rabbit anti-TPH2 primary antibodies (Millipore, ABN60) diluted 1:1000. After incubation in the primary antibody carrier solution, slices were rinsed 4 × 5 min with PBST, and then incubated in carrier solution containing goat anti-rabbit Alexa Fluor 647 at 1:500 (ThermoFisher Scientific, A-21244) and Neurotrace 435 at 1:100 for 2 hr at room temperature. Slices were then rinsed 2 × 5 min in PBST and 2 × 5 min in PBS before mounting with VectaShield mounting media (Vector Labs, H-1000).

## Fluorescent in situ Hybridization (*FISH*)

Animals were deeply anesthetized with isoflurane before decapitation. Brains then were rapidly removed, frozen on dry ice, and embedded in tissue freezing media (Tissue-Tek O.C.T. compound). 20 μm coronal sections were prepared on a cryostat (Leica Biosystems, CM1950) and mounted onto SuperFrost Plus glass slides (VWR) at 100 μm intervals between slices in a set. Slices were rapidly refrozen once mounted, and stored at −80°C before staining. Multiplexed fluorescent in situ hybridization was performed using the ACDBio RNAscope V1 reagents and protocols. Briefly, slices were fixed in 4% paraformaldehyde at 4°C for 15 min and dehydrated through washing steps in ethanol at increasing concentrations (50%, 70%, 100%) before protease digestion (Protease III, 10 min at room temperature). Probes and amplification/detection reagents were applied to the tissue sections and incubated under conditions stated in the V1 detection protocol provided by ACDBio. Sections were counterstained using DAPI provided in the V1 detection reagent kits and mounted in ProLong Gold mounting media (ThermoFisher Scientific P36934). Single-plane tiled images covering the DRN were scanned using a confocal microscope (Leica SP8X, 1.40 NA 63X magnification oil immersion objective). Tile merging was performing in the Leica LAS X software, with a 10% tile overlap and statistical blending.

## *FISH* image analysis

Confocal images were first processed in Fiji. A custom macro was written to convert Leica image files (.lif format) into TIFF format for downstream analysis steps. Channels in the merged tile images were processed separately. For each channel, a median filter (one pixel radius) was applied before images were cropped and downsampled by a factor of 2 with bicubic interpolation. A rolling ball background subtraction (50 pixel radius) was then applied before images for each channel was saved as a separate 8-bit TIFF file. A pixel classifier run in Ilastik (version 1.2.2) was applied on the output TIFF files to generate probability maps of *B19N*-expressing cell bodies for automated segmentation of rabies-infected cells. Cell segmentation and RNAscope puncta quantification was performed using a custom pipeline in CellProfiler (version 3.1.5). Puncta counts were exported as TSV files for further analysis and data visualization using custom scripts written in R and MATLAB.

## Acknowledgements

We thank the Dymecki (HMS) and Lowell (BIDMC) Labs for generously providing transgenic mouse lines; the HMS ICCB Single Cell Core for assistance with scRNA-seq experiments on the InDrop platform; S Hrvatin and A Nagy (Greenberg Lab, HMS) for advice and help with scRNA-seq protocols and analysis pipelines; BK Lim (UCSD) and IR Wickersham (MIT) for advice and reagents for rabies virus production; the Bauer Core Facility at Harvard University for sequencing support; the HMS Neurobiology Imaging Facility for confocal microscopy support (P30 NS072030); and members of the Sabatini Lab for helpful discussions. We also thank our undergraduate students M Henson, A Mustoe, and L Chung for technical assistance, J Levasseur for genotyping and animal husbandry, and L Worth for administrative assistance. This work was supported by funding from the Howard Hughes Medical Institute (BLS), National Institutes of Health (R01 MH100568 and R01 NS103226 to BLS), a Harvard Brain Initiative Bipolar Disorder Seed Grant (BLS), a Samsung Scholarship (MH), the Lefler Center for the Study of Neurodegenerative Disorders (MH), the HMS Department of Neurobiology Graduate Fellowship (KWH), and the HMS Stuart HQ and Victoria Quan Fellowship in Neurobiology (KWH).

## Additional information

### Funding

| Funder | Grant reference number | Author |
| --- | --- | --- |
| National Institute of Neurological Disorders and Stroke | NS103226 | Bernardo L Sabatini |
| Howard Hughes Medical Institute | | Bernardo L Sabatini |

| National Institute of Mental Health | MH100568 | Bernardo L Sabatini |
|---|---|---|
| Harvard Brain Initiative | Bipolar Disorder Seed Grant | Bernardo L Sabatini |
| Samsung | Samsung Scholarship | Minsuk Hyun |
| Harvard Medical School | Lefler Center for the Study of Neurodegenerative Disorder | Minsuk Hyun |
| Department of Neurobiology, Harvard Medical School | Graduate Fellowship | Kee Wui Huang |
| Department of Neurobiology, Harvard Medical School | Stuart HQ and Victoria Quan Fellowship in Neurobiology | Kee Wui Huang |

The funders had no role in study design, data collection and interpretation, or the decision to submit the work for publication.

## Author contributions

Kee Wui Huang, Conceptualization, Resources, Data curation, Software, Formal analysis, Supervision, Funding acquisition, Validation, Investigation, Visualization, Methodology, Writing—original draft, Project administration, Writing—review and editing; Nicole E Ochandarena, Formal analysis, Investigation, Writing—review and editing; Adrienne C Philson, Formal analysis, Investigation, Methodology; Minsuk Hyun, Investigation, Methodology; Jaclyn E Birnbaum, Formal analysis, Investigation, Visualization; Marcelo Cicconet, Software, Formal analysis, Visualization, Methodology; Bernardo L Sabatini, Conceptualization, Funding acquisition, Methodology, Writing—original draft, Project administration, Writing—review and editing

## Author ORCIDs

Kee Wui Huang (iD) https://orcid.org/0000-0003-2265-4550
Bernardo L Sabatini (iD) https://orcid.org/0000-0003-0095-9177

## Ethics

Animal experimentation: All procedures were performed in accordance with protocols approved by the Harvard Standing Committee on Animal Care following guidelines described in the US National Institutes of Health Guide for the Care and Use of Laboratory Animals (HMS IACUC protocol #IS00000571). All surgery was performed under isoflurane anesthesia.

## Decision letter and Author response

Decision letter https://doi.org/10.7554/eLife.46464.031
Author response https://doi.org/10.7554/eLife.46464.032

# Additional files

## Supplementary files

• Supplementary file 1. Genes for identification of major cell types. Several examples of genes used to identify each of the major cell types/classes are listed. *Slc17a7* (VGluT1) is typically used to identify glutamatergic neurons but was not detected in our scRNA-seq dataset, and was therefore excluded from this list.
DOI: https://doi.org/10.7554/eLife.46464.023

• Supplementary file 2. Genes for spatial correlation analysis. Table of genes and the corresponding experiment IDs from the Allen Brain Atlas that were used for spatial correlation analysis to infer the distribution of 5-HT neuron subtypes. Images, referenced in the table by the image numbers in the experiment image series, were manually matched to a corresponding image in the coronal reference that consisted of 12 sections spanning −3.80 mm to −4.90 mm along the anterior-posterior axis (zeroed at Bregma). Missing data (e.g. no image, damaged section), is denoted with a "-" and

assigned a NaN value. Sections containing data for the same gene from different experiments were averaged to obtain a single entry for each gene.
DOI: https://doi.org/10.7554/eLife.46464.024

• Transparent reporting form
DOI: https://doi.org/10.7554/eLife.46464.025

## Data availability

The sequencing datasets generated in this study are available on the NCBI Gene Expression Omnibus (accession number: GSE134163). R data files containing the processed and annotated scRNA-seq data in the form of Seurat objects are also available on the Harvard Dataverse (https://doi.org/10.7910/DVN/QB5CC8).

The following datasets were generated:

| Author(s) | Year | Dataset title | Dataset URL | Database and Identifier |
|---|---|---|---|---|
| Kee Wui Huang, Bernardo L Sabatini | 2019 | scRNA-seq_huang2019 | https://doi.org/10.7910/DVN/QB5CC8 | Harvard Dataverse, 10.7910/DVN/QB5CC8 |
| Huang KW, Sabatini BL | 2019 | Molecular and anatomical organization of the dorsal raphe nucleus | https://www.ncbi.nlm.nih.gov/geo/query/acc.cgi?acc=GSE134163 | NCBI Gene Expression Omnibus, GSE134163 |

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
