## [Decision Letter]

Thank you for submitting your article "Molecular and anatomical organization of the dorsal raphe nucleus" for consideration by *eLife*. Your article has been reviewed by three peer reviewers, and the evaluation has been overseen by a Reviewing Editor and Eve Marder as the Senior Editor. The following individuals involved in review of your submission have agreed to reveal their identity: Tibor Harkany (Reviewer #1); Idoia Quintana-Urzainqui (Reviewer #3).

The reviewers have discussed the reviews with one another and the Reviewing Editor has drafted this decision to help you prepare a revised submission.

Summary:

This is a comprehensive, well executed and interesting study addressing a timely and important question. The raphe nuclei are implicated in many neuropsychiatric and neurological disorders and they hold one third of all serotonergic neurons of the brain. However, the structure and function of the region has been difficult to study given its extraordinary heterogeneity in neurochemical composition and anatomical organization. We are in need, therefore, of single cell datasets and studies like this one. In their manuscript "Molecular and anatomical organization of the dorsal raphe nucleus", Huang and colleagues shed light on the complex anatomy and organization of the dorsal raphe nucleus (DRN) in the adult mouse. By performing single cell RNA-seq in dissected DRN region they produced an interesting dataset which encompasses neural and non-neural cell types present in and around DRN. Based on this transcriptomic data they identify 5 subtypes of 5-HT neurons and comprehensively study their anatomical and molecular characteristics. By analyzing the distribution of genes identified as differentially expressed on each of the putative subtypes they develop a map of their anatomical localization within the DRN. Then, they perform a series of experiments to explore whether 5-HT raphe neurons projecting to known targets (basal ganglia, cortex or thalamus) correspond to any of the subtypes identified. They show that the DR is composed of at least 5 distinct 5-HT neuronal types. In addition, the authors demonstrate that at least 2 distinct 5-HT neuronal types project to the striatum. Finally, they validate that a *Pdyn*-expressing 5-HT neurons (likely belonging to subtype III) strongly and specifically innervate various basal ganglia structures.

Essential revisions:

The reviewers have very positive comments about the study, but especially because this paper will be relied upon as a critical reference in the field, more validation is required for the existence of the 5 subtypes, particularly given the relatively low percentage of genes identified per cell with the single-cell RNA-seq. To achieve this, two approaches are indicated.

1) Additional *FISH* will be required to prove that the 5 claimed subtypes are correct and separate (in particular, the two very similar cell types 5-HT-I and II). Adding a reasonable quantitative would be adequate (e.g. 5-HT-IV x% out of all serotonergic neurons in a particular DRN subregion), and would be a good fit to Figure 4B. Preferably the authors will use at least one or two more specific markers for each cell subtype, showing similar anatomical distribution throughout the DRN and/or co-localization if possible.

2) Sten Linnarsson has recently published a scRNA-seq paper in Cell (Zeisel et al., 2018) in which they find TPH2^+^ neurons. Those data were produced with deeper sequencing. It would elevate the importance of the findings in this manuscript considerably if the authors could use a bioinformatic comparison of their own cells with this publicly available database. First, they should mix together their own cells and those from the Linnarsson database and see if iterative machine learning splits them into the two original clusters; if so, this may then require further sequencing. If not, they then might go deeper with a larger number of cells to justify the existence of the 5 subgroups. The above pipeline could also be informative in retaining the "peptidergic" cluster. There is concern that the authors may be losing low-level expressed neurotransmitter synthesis genes.

[Editors' note: further revisions were requested prior to acceptance, as described below.]

Thank you for submitting your article "Molecular and anatomical organization of the dorsal raphe nucleus" for consideration by *eLife*. Your article has been reviewed by three peer reviewers, and the evaluation has been overseen by a Reviewing Editor and Eve Marder as the Senior Editor. The following individuals involved in review of your submission have agreed to reveal their identity: Idoia Quintana-Urzainqui (Reviewer #3); Martin Häring (Reviewer #4).

The reviewers have discussed the reviews with one another and the Reviewing Editor has drafted this decision to help you prepare a revised submission.

Summary:

This is a comprehensive, well executed and interesting study addressing a timely and important question. By performing single cell RNA-seq in dissected DRN region they produced an interesting dataset which encompasses neural and non-neural cell types present in and around DRN. Based on this transcriptomic data they identify 5 subtypes of 5-HT neurons and comprehensively study their anatomical and molecular characteristics. By analyzing the distribution of genes identified as differentially expressed on each of the putative subtypes they develop a map of their anatomical localization within the DRN. Then, they perform a series of experiments to explore whether 5-HT raphe neurons projecting to known targets (basal ganglia, cortex or thalamus) correspond to any of the subtypes identified. The work is expected to be widely used as a reference for this important brain region.

The reviewers are generally enthusiastic about this paper, and the substantial revisions to the original manuscript. They do, however, have some remaining concerns that we feel can be easily addressed.

We suggest a change in the Figure 4—figure supplement 2. The Z-Score has the disadvantage that it only shows relative expression levels (colour code). In respect to expression differences this can be misleading. Thus, if you compare several different cell types (Figure 3B) the expression (here *Tph2* and *Slc22a3*) are quite uniform among the serotonergic neurons. If, however, the observed number of clusters is smaller (focus on serotonergic cells; Figure 4A), than the small expression differences suddenly appear huge for these two genes. Moreover, the authors state in the figure legend for Figure 4—figure supplement 2 that "…*Prkcq^+^;Trh^-^* 5-HT neurons are putative 5-HT-II neurons.". This might be correct but has to be shown in another mean beside Z-score as both genes are shown to be expressed also in Cluster 2 (slightly lower compared to Cluster 1) in Figure 3B and even in 4A.

The reviewers agree that the paper would be stronger if the authors would add staining for *Nfix* or *Slc6a1* (Cluster 1; Figure 4) and *Mkl2* (Cluster 2 enriched), although we will leave this up to the authors. If no new stainings are added, however, the presentation of Z-scores should be changed to dotblot (or maybe barblot) format, as seen in other publications (e.g. Kupari et al., 2019, Cell Rep.). Dotblots allow the visualization of the percentage of cells expressing the gene in the separate clusters in addition to relative expression levels (colour code), while the bar blot approach shows the expression level of a gene for each single cell directly.

This presentation will allow the reader to determine whether expression differences are due to generally lower levels in the two clusters or to higher percentage of cells lacking the gene in Cluster 2. The Z-score does not allow this, and the TPH2 and *Slc22a3* results are examples of how the Z-Score can be misleading (Figure 3B and 4A).

---

## [Author Response]

Essential revisions:The reviewers have very positive comments about the study, but especially because this paper will be relied upon as a critical reference in the field, more validation is required for the existence of the 5 subtypes, particularly given the relatively low percentage of genes identified per cell with the single-cell RNA-seq. To achieve this, two approaches are indicated.1) Additional FISH will be required to prove that the 5 claimed subtypes are correct and separate (in particular, the two very similar cell types 5-HT-I and II). Adding a reasonable quantitative would be adequate (e.g. 5-HT-IV x% out of all serotonergic neurons in a particular DRN subregion), and would be a good fit to Figure 4B. Preferably the authors will use at least one or two more specific markers for each cell subtype, showing similar anatomical distribution throughout the DRN and/or co-localization if possible.

We thank the reviewers for their positive comments and thoughtful feedback on our manuscript.

To address the concerns raised in Essential Revision #1, we have included additional data from *FISH* experiments to validate the differential expression of several subtype enriched genes and the results of our spatial correlation analysis. These results have been added to Figure 4G-I and an additional supplementary figure (Figure 4—figure supplement 2 in our revised manuscript). For the quantitation of the proportion of 5-HT neurons expressing subtype-enriched genes in each DRN subregion, we have instead chosen to present this data as probability distributions of 5-HT neurons with detectable levels of the transcripts of interest (Figure 4I, Figure 4—figure supplement 2E-F). We have refrained from performing this analysis since the boundaries between DRN subregions are poorly defined. This is further supported by our findings that 5-HT neurons expressing subtype enriched transcripts are not confined to well-defined DRN subregions, and are often found to be spatially intermingled despite having different peak density locations. This is consistent with the results of our spatial correlation analysis (Figure 5).

With regard to the validation of the 5-HT neuron subtypes by *FISH* experiments staining for subtype-specific marker genes, we wish to clarify the distinction between subtype enriched genes and subtype marker genes. Subtype marker genes are a subset of subtype enriched genes that are exclusively expressed in a given subtype, whereas subtype enriched genes may be expressed in other subtypes but at different levels that are still informative for discriminating between the two groups. While marker genes are indeed useful for validating a given cluster/subtype, given the strong expectation of a binary expression pattern, an inability to detect marker genes despite finding subtype enriched genes may result from technical limitations of high-throughput scRNA-seq methods (e.g. detection threshold, transient expression). Given the trade-off between sequencing depth and the number of cells sampled when using droplet-based scRNA-seq techniques, we have refrained from suggesting a list of marker genes for all subtypes and instead report the enriched genes that we have sufficient confidence in.

With regard to the validation of the separation of cells in the 5-HT-I and 5-HT-II clusters, we have provided additional *FISH* data in Figure 4—figure supplement 2D in which we found that *Trh^+^* cells are a subset *Prkcq*^+^ cells, as predicted from our sequencing data. We also found enriched genes for both clusters despite their similarities, and that cells in both clusters had comparable read depth and gene detection rates (see Author response image 1). Therefore, the separation of 5-HT-I and 5-HT-II is unlikely to be due to insufficient sequencing of cells in either cluster, although methods that can achieve deeper sequencing will likely resolve further separation of the two.

**Author response image 1. respfig1:** UMI count and gene detection rate for 5-HT neuron clusters. Violin plots for UMI counts (nUMI, left) and gene detection rate (nGene, right) for each of the 5 5-HT neuron clusters in our scRNA-seq dataset. Columns represent each subtype, and individual points represent single cells.

2) Sten Linnarsson has recently published a scRNA-seq paper in Cell (Zeisel et al., 2018) in which they find TPH2^+^ neurons. Those data were produced with quite deeper sequencing. It would elevate the importance of the findings in this manuscript considerably if the authors could use a bioinformatic comparison of their own cells with this publicly available database. First, they should mix together their own cells and those from the Linnarsson database and see if iterative machine learning splits them into the two original clusters; if so, this may then require further sequencing. If not, they then might go deeper with a larger number of cells to justify the existence of the 5 subgroups. The above pipeline could also be informative in retaining the "peptidergic" cluster. There is concern that the authors may be losing low-level expressed neurotransmitter synthesis genes.

We agree with the reviewers that a comparison of our dataset with the dataset from Zeisel et al. (2018), which is one of the largest scRNA-seq datasets that contains DRN 5-HT neurons, will be very valuable. As such, we have performed additional analyses incorporating the 5-HT neurons in the Zeisel 2018 dataset and present our results and findings below. The R notebooks documenting the analyses are available on GitHub (Sabatini, 2019; copy archived at https://github.com/elifesciences-publications/DRN_scRNAseq_Huang2019).

The Zeisel et al. (2018) dataset contains 5-HT neurons from many other raphe nuclei besides the DRN. The level of clustering that we have performed on the 5-HT neurons in our dataset also goes further than described in the Zeisel et al., 2018, which did not subdivide the HBSER1 cluster into additional subclusters/subtypes. Zeisel et al. noted in their data portal that there is additional heterogeneity in expression of *Nos1*, which is enriched in 5-HT-III in our dataset, within the HBSER1 cluster. Furthermore, the HBSER1 cluster in the Zeisel 2018 dataset, which their spatial correlation analysis had predicted to localize to the DRN and/or MRN, contains fewer cells than our dataset (118 vs. 704 cells).

Additionally, we note that neurons in the Zeisel 2018 dataset that were assigned to the MEGLU14 cluster, which are likely to correspond to the peptidergic neurons in our dataset, showed near-zero expression of *Slc17a8* close to the noise floor. As such, we suspect that there was a potential error in the neurotransmitter calling for this cluster, rather than an inability to detect transcripts of *Slc17a8* or other neurotransmitter synthesis genes in our peptidergic neuron cluster.

To compare the DRN 5-HT neurons across both datasets, we first analyzed the 5-HT neurons in the Zeisel 2018 dataset to isolate DRN 5-HT neurons in their dataset. Clustering cells from the HBSER1-HBSER5 clusters independent of their pre-existing cluster identity assignments resulted in 6 clusters that differed from the assignment of 5 clusters by Zeisel et al. (Author response image 2). We identified these clusters by examining the cluster-enriched genes and found that these 6 clusters correspond to the different subsets of 5-HT neurons from different embryonic rhombomeres as described by Okaty et al. (2015). Clusters in the Zeisel 2018 dataset included most of the 5-HT neuron groups described in Okaty et al. (2015) with the exception of R3-derived 5-HT neurons. We speculate that this is due to the relatively low sampling of 5-HT neurons in the Zeisel 2018 dataset relative to the number of 5-HT neurons in the CNS, and the low abundance of R3 neurons in the MRN.

**Author response image 2. respfig2:** DRN 5-HT neurons are a subset of 5-HT neurons in the dataset from Zeisel et al. (2018). 5-HT neurons from the HBSER1-HBSER5 clusters in the dataset accompanying the publication by Zeisel et al., 2018, were isolated and clustered independently. Left: UMAP plot of 5-HT neurons in the Zeisel 2018 dataset color-coded by cluster assignments from graph-based clustering (Seurat, v. 2.3.1). Clusters identities were assigned by comparing cluster-enriched genes with genes known to be enriched in different 5-HT neuron subsets as described in Okaty et al., 2015. The separation of the “R1 (DRN)” cluster from “R1 (medial DRN)” and “R1 (lateral DRN)” clusters is largely driven by differences in expression of housekeeping genes such as ribosomal genes, and are likely to be technical in nature similar to batch effects (data not shown). Right: UMAP plot of 5-HT neurons in the Zeisel 2018 dataset color-coded by the original cluster assignments. Cluster assignments from the originating publication differ from the clusters that we have identified independently using graph-based clustering methods.

For further comparison of the single-cell RNA sequencing datasets with the pooled cell RNA-seq samples from Okaty et al. (2015), we generated simulated bulk RNA-seq data from our dataset and the Zeisel 2018 dataset by summing the counts for each gene across all cells in a given cluster. All cells in our dataset were combined into one simulated pool since our dataset consisted of only R1-derived DRN neurons. The MDS plot (Author response image 3) showed that the pooled RNA-seq samples from Okaty et al. (2015) and the simulated bulk RNA-seq samples separated along dimension 1, which is likely to be driven primarily by technical differences (e.g. platforms used for cell isolation, mRNA capture, library preparation). However, corresponding clusters across datasets showed similarities in their position along dimension 2.

**Author response image 3. respfig3:** Clusters identified from Zeisel 2018 dataset correspond to 5-HT neurons originating from different rhombomeres. Multidimensional scaling (MDS) plot of pooled samples from Okaty et al. (2015) and simulated bulk RNA-seq samples from our dataset (Huang_5HT) and clusters we identified in the Zeisel 2018 dataset (Zeisel_R1DR to Zeisel_R6P). Separation of samples along dimension 1 corresponds primarily to the dataset of origin, and is likely to be driven by technical effects. Samples that are biologically similar, i.e. containing cells from similar structures or developmental origin, are closer in position along dimension 2. The MDS plot was generated using the edgeR package.

The reviewers argued that a separation of cells from our dataset and the Zeisel 2018 dataset is indicative of insufficient sequencing depth in our dataset compared to the Zeisel 2018 dataset. However, we disagree with this claim, and provide evidence against it from our analysis of a merged object combining the DRN 5-HT neurons from the Zeisel 2018 dataset with the 5-HT neurons in our dataset.

The merging of independent datasets is likely to give rise to separate clusters due to batch effects and other technical sources of variation that are more indicative of differences in protocols and scRNA-seq platforms rather than a difference in sequencing depth. We can obtain this separation of cells from the Zeisel 2018 dataset from our dataset when visualizing the data in a UMAP plot generating using the first 11 principal components (PCs) of the merged dataset (Author response image 4). The plot also shows a separation of cells from different batches within our dataset, which motivated the use of CCA-based dataset alignment for the analyses in our manuscript. Inspection of features in the data that include the number of UMIs (nUMI) or genes (nGene) detected per cell and PC scores for each cell in the 2D UMAP embedding showed that PCs 1-3 correlated better with the separation of the cells by dataset/batch than either nUMI or nGene (Author response image 6). Examining the genes with the highest absolute weights in PCs 1-3 showed that these are dominated by ubiquitously expressed genes such as ribosomal and mitochondrial genes (Author response image 5), suggesting that these effects are largely technical in nature.

**Author response image 4. respfig4:** DRN 5-HT neurons in merged dataset separate by study of origin and batch number. DRN 5-HT neurons from the Zeisel 2018 dataset were merged with 5-HT neurons in our dataset into a single object that was analyzed in Seurat. UMAP plots were generated using the first 11 principal components (PCs). Cells are color-coded by the batch of origin. Red: Huang dataset batch 1. Blue: Huang dataset batch 2. Gray: Zeisel 2018 dataset. Batch information within the Zeisel 2018 dataset was not available.

**Author response image 5. respfig5:** Separation of cells by dataset and batch can be attributed to the first 3 principal components. UMAP plots of different data features, such as PC loadings and UMI counts per cell where each cell is color-coded by its score for a given feature. PCs 1-3 correlate better with the separation of cells by batch and dataset compared to the UMI count (nUMI) or number of genes (nGene) detected per cell.

**Author response image 6. respfig6:** Batch effects are largely driven by differences in ubiquitously expressed genes. Heatmaps showing the scaled expression (yellow: high, purple: low) of the genes with the highest weights in PCs 1-3 in cells at both extremes of the PC scores. Ubiquitously expressed genes such as ribosomal and mitochondrial genes have the highest weights in these PCs, which correlate with the separation of cells by dataset and batch.

Despite the separation of the cells by dataset in Author response image 4, we noted that the cluster of cells from the Zeisel 2018 dataset contained at least 4 subdivisions that can also be seen in the UMAP plot. Differences in the expression of subtype-enriched genes in these 4 subdivisions (Author response image 7) suggested that these subdivisions correspond to the 5-HT neuron subtypes I-IV that we identified in our dataset. 5-HT-V neurons were not found in this subset of cells from the Zeisel 2018 dataset due to two main factors: (i) the Zeisel 2018 dataset sampled relatively few 5-HT neurons and may not have captured a sufficient number of 5-HT-V neurons since these form a very rare population, and (ii) the small number of *Met*^+^ 5-HT-V cells that were present in the HBSER1-HBSER5 clusters of the Zeisel 2018 dataset clustered with the R1/R2 MRN cluster (data not shown), which was expected given the transcriptional similarities between the caudal DRN and MRN that have been previously described.

**Author response image 7. respfig7:** The Zeisel 2018 dataset 5-HT neuron cluster contains subdivisions corresponding to 5-HT neuron subtypes. UMAP plots showing the expression of 5-HT neuron marker genes and several subtype enriched genes. Cells are color-coded by their log-normalized expression of the respective genes. 4 subdivisions can be seen within the cluster of cells from the Zeisel 2018 dataset on the right-hand side of each UMAP plot. Each subdivision is enriched in a different set of subtype enriched genes and are likely to correspond to clusters of 5-HT neuron subtypes I-IV. 5-HT-V neurons were not found in this subset of the Zeisel 2018 dataset.

Several methods have been used by published studies to mitigate the separation of clusters by batch effects. One approach that is similar to the method used by Tasic et al. (2018) is the removal of principal components that correspond to or correlate with quality control metrics, which would be PCs 1-3 in the merged dataset that we have analyzed here. Exclusion of PCs 1-3 from the calculation of the 2-dimensional UMAP embedding abolished the separation of cells by dataset and batch (Author response image 8). Similar results were obtained when using CCA-based dataset alignment techniques that are available in the Seurat package (Author response image 9). Cells from the Zeisel 2018 dataset were found to co-cluster with the corresponding 5-HT neuron subtype clusters from our dataset (R1DR_lateral with 5-HT-I/II, and R1DR_medial with 5-HT-III/IV) using either method. These results suggest that the DRN 5-HT neurons in the Zeisel 2018 dataset exhibit additional heterogeneity in agreement with the subtypes that we have described in our study. However, inclusion of the small number of additional cells did not provide any significant improvements in resolving additional clusters beyond what we have already described.

**Author response image 8. respfig8:** Cells from different datasets co-cluster after exclusion of PCs related to batch effects. Exclusion of PCs 1-3 when generating a UMAP plot of the merged dataset removes the separation of cells by dataset and batch. Left: UMAP plot of the merged dataset, with cells color-coded by the dataset of origin. Right: UMAP plot of the merged dataset, with cells color-coded by the cluster identities assigned during separate analysis of each dataset. Cells from the Zeisel 2018 dataset co-cluster with the corresponding subtypes from our dataset.

**Author response image 9. respfig9:** Cells from different datasets co-cluster when merged using CCA-based dataset alignment. CCA-based dataset alignment (Butler et al., 2018) was performed on the merged dataset prior to 2-dimensional UMAP embedding. UMAP plots were generated using the first 7 aligned CVs. Left: UMAP plot of the merged dataset, with cells color-coded by the dataset of origin. Right: UMAP plot of the merged dataset, with cells color-coded by the cluster identities assigned during separate analysis of each dataset.

[Editors' note: further revisions were requested prior to acceptance, as described below.]

The reviewers are generally enthusiastic about this paper, and the substantial revisions to the original manuscript. They do however, have some remaining concerns that we feel can be easily addressed.We suggest a change in the Figure 4—figure supplement 2. The Z-Score has the disadvantage that it only shows relative expression levels (colour code). In respect to expression differences this can be misleading. Thus, if you compare several different cell types (Figure 3B) the expression (here Tph2 and Slc22a3) are quite uniform among the serotonergic neurons. If, however, the observed number of clusters is smaller (focus on serotonergic cells; Figure 4A), than the small expression differences suddenly appear huge for these two genes. Moreover, the authors state in the figure legend for Figure 4—figure supplement 2 that "…Prkcq^+^;Trh^-^ 5-HT neurons are putative 5-HT-II neurons.". This might be correct but has to be shown in another mean beside Z-score as both genes are shown to be expressed also in Cluster 2 (slightly lower compared to Cluster 1) in Figure 3B and even in 4A.The reviewers agree that the paper would be stronger if the authors would add staining for Nfix or Slc6a1 (Cluster 1; Figure 4) and Mkl2 (Cluster 2 enriched), although we will leave this up to the authors. If no new stainings are added, however, the presentation of Z-scores should be changed to dotblot (or maybe barblot) format, as seen in other publications (e.g. Kupari et al., 2019, Cell Rep.). Dotblots allow the visualization of the percentage of cells expressing the gene in the separate clusters in addition to relative expression levels (colour code), while the bar blot approach shows the expression level of a gene for each single cell directly.

We have included dot plots showing expression of the genes in Figure 4A as log-transformed scaled UMI counts instead of Z-scored expression. These plots have been included in new supplements to both Figure 3 (Figure 3—figure supplement 1) and Figure 4 (now Figure 4—figure supplement 1). However, we note that these plots emphasize highly expressed genes over differentially expressed genes that may have fewer transcripts and which are more apparent in the heatmaps showing Z-scored relative expression.

This presentation will allow the reader to determine whether expression differences are due to generally lower levels in the two clusters or to higher percentage of cells lacking the gene in Cluster 2. The Z-score does not allow this, and the TPH2 and Slc22a3 results are examples of how the Z-Score can be misleading (Figure 3B and 4A).

We respectfully disagree with our reviewers regarding their concern that relative expression levels presented as Z-scores are “misleading”. Relative differences in expression across multiple genes are important for distinguishing cell types, and are collectively more informative than single “marker” genes that exhibit strongly bimodal or binarized patterns of expression. The use of Z-scores in Figure 4, which takes into account the variance of expression of each gene, is intended to report these differences without biasing the reader towards genes with either “binary” expression patterns or high expression, which are emphasized by the dot plot and suffer less from the effects of drop-outs characteristic of many scRNA-seq methods. Drop-outs cause these low copy number transcripts to appear has having a much lower fraction of cells expressing in the dot plot. The relative differences in the expression of highly expressed genes that the reviewers pointed to from Figure 3 are seen more easily with the centered Z-scored expression when making comparisons *within* the 5-HT neuron group (Figure 4), rather than across the full neuronal population (Figure 3). We also point out that we have observed with *FISH* the high but relatively lower expression of *Slc6a4* in 5-HT-V compared to the other four subtypes (see images in Figure 4—figure supplement 3C of the revised manuscript). This observation from our *FISH* experiments supports the biological rather than purely technical nature of these differences that are highlighted by the Z-scores.